# Unifying Re-Identification, Attribute Inference, and Data Reconstruction Risks in Differential Privacy

**Bogdan Kulynych**
Lausanne University Hospital

**Juan Felipe Gomez**
Harvard University

**Georgios Kaissis**
Google DeepMind

**Jamie Hayes**
Google DeepMind

**Borja Balle**
Google DeepMind

**Flavio P. Calmon**
Harvard University

**Jean Louis Raisaro**
Lausanne University Hospital
University of Lausanne

## Abstract

Differentially private (DP) mechanisms are difficult to interpret and calibrate because existing methods for mapping standard privacy parameters to concrete privacy risks—re-identification, attribute inference, and data reconstruction—are both overly pessimistic and inconsistent. In this work, we use the hypothesis-testing interpretation of DP ($f$-DP), and determine that bounds on attack success can take the same unified form across re-identification, attribute inference, and data reconstruction risks. Our unified bounds are (1) consistent across a multitude of attack settings, and (2) tunable, enabling practitioners to evaluate risk with respect to arbitrary, including worst-case, levels of baseline risk. Empirically, our results are tighter than prior methods using $\varepsilon$-DP, Rényi DP, and concentrated DP. As a result, calibrating noise using our bounds can reduce the required noise by 20% at the same risk level, which yields, e.g., an accuracy increase from 52% to 70% in a text classification task. Overall, this unifying perspective provides a principled framework for interpreting and calibrating the degree of protection in DP against specific levels of re-identification, attribute inference, or data reconstruction risk.

## 1 Introduction

Releases of models and statistics derived from personal data—such as de-identified and synthetic data releases, outcomes of statistical analyses, and machine-learning models—can reveal information about individuals in the data. These releases can be classified into *record-level* (releasing de-identified or obfuscated records) and *aggregate* (releasing statistical properties and models) (Gadotti et al., 2024). The disclosure of such information incurs different types of privacy risks, which are legislated differently across jurisdictions and regulatory frameworks.

In record-level releases, information leakage can occur via *singling out* of an individual (Cohen and Nissim, 2020), e.g., determining that there is only one Colombian woman fluent in Mandarin over the age of 60 in the dataset. When combined with external information, singling out can result in a person's *re-identification.* Re-identification is harmful when it enables inference of previously unknown personal attributes by an attacker, e.g., disease status or voting record (Sweeney, 2002). Although in aggregate releases there is no one-to-one mapping between the released information and individuals, sensitive information can still be inferred via *attribute inference* or *data reconstruction attacks* (Fredrikson et al., 2015; Yeom et al., 2018; Balle et al., 2022), in which the adversary could,

39th Conference on Neural Information Processing Systems (NeurIPS 2025).

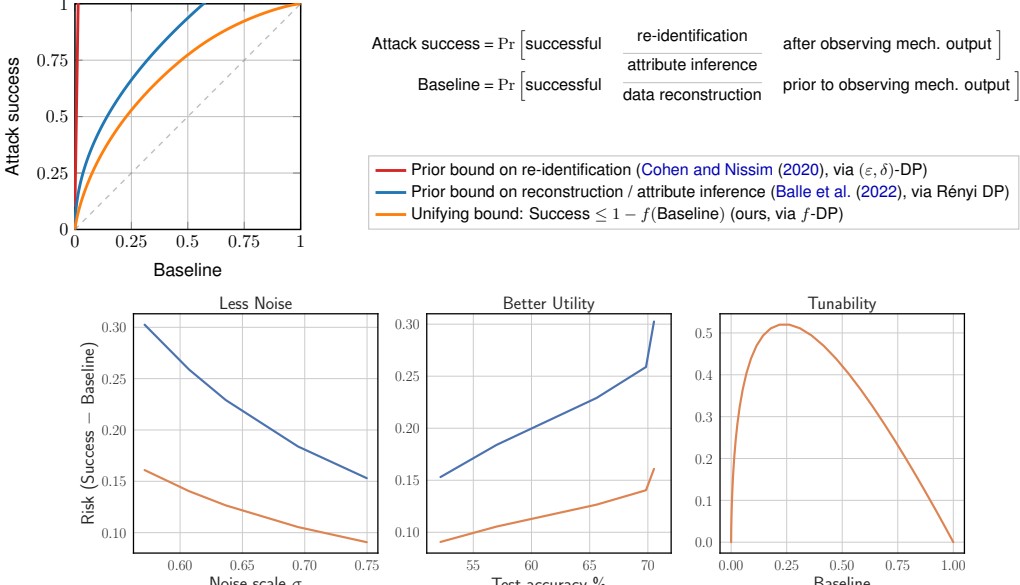

Figure 1: Our results offer a unified and more precise way to interpret and calibrate DP mechanisms in terms of re-identification, attribute inference, and data reconstruction risks. *Top:* The success of all these attacks cannot be higher than the power of the worst-case membership inference attack, which we immediately obtain from the $f$ function in the decision-theoretic characterization of privacy mechanisms—$f$-DP. *Bottom left:* Our results enable to add $\approx 20\%$ less noise at any given level of risk compared to using prior methods (see Section 4 for details). *Bottom middle:* Less noise translates into $\approx 18$pp improved task accuracy (shown: DP-SGD for sentiment classification with GPT-2). *Bottom right:* The unifying risk measure is tunable—we can either estimate post-release risk for any given level of baseline risk, or measure the worst-case risk (shown: US 2020 Census state-level data).

e.g., inspect the confidence outputs of a predictive model to infer one or more unknown attributes of an individual. These notions of risk are interpretable and appear in data-protection guidelines such as ISO standards (ISO/IEC 27559:2022), or guidelines from the European Medicines Agency (European Medicines Agency, 2018).

Differential privacy (DP) (Dwork et al., 2014) is a standard tool for controlling information leakage in record-level and aggregate data disclosures. The goal of DP mechanisms is to mask the contribution of each individual in the release via noise injection and randomization. One of the core challenges in applying DP in practice is its interpretation (Franzen et al., 2022; Nanayakkara et al., 2023), as DP parameters do not trivially bound concrete risks such as re-identification, attribute inference, and data reconstruction. In the standard practice, information leakage in DP is quantified by two parameters: $\varepsilon$, with values closer to zero meaning better privacy, and $\delta$, which should be kept cryptographically small (Vadhan, 2017).

There are few established guidelines for selecting DP parameters $(\varepsilon, \delta)$, and often contradictory standards are adopted in practice. For example, heuristics such as $\varepsilon < 10$ are common in deep learning (Ponomareva et al., 2023). When translated to concrete privacy risks, such choices can result in vacuous privacy guarantees: in theory, $\varepsilon \approx 10$ can yield a more than 99% worst-case attribute inference accuracy.[1] This poor performance *is not* due to inherent limitations of DP itself—instead, it is a reflection of the imprecise nature of existing methods that convert between DP guarantees and notions of privacy risk. In fact, empirical evidence shows that even $\varepsilon > 10$ still prevents practical reconstruction attacks (Ziller et al., 2024; Hayes et al., 2024).

In this paper, we use the decision-theoretic characterization of DP known as $f$-DP (Dong et al., 2022) to unify and improve existing conversion bounds between DP and different notions of privacy risk. We work under the *strong-adversary threat model*, in which the adversary has access to the entire dataset

---

[1]Attack accuracy is bounded by $\frac{e^\varepsilon + \delta}{e^\varepsilon + 1}$ (Kairouz et al., 2015; Humphries et al., 2023; Salem et al., 2023).

except for one element, and has a certain *prior distribution* over the remaining element. Inspired by Cohen and Nissim (2020), we characterize the attack *success probability* of the adversary after observing the output of a DP mechanism as a function of the attack's *baseline probability* without the DP output. Surprisingly, we find that under the strong threat model, we can simultaneously bound the success probability of re-identification, attribute inference, and reconstruction attacks at once, using the same expression. Importantly, we find that not only this is a convenient way to analyze multiple kinds of risk at once, but also it provides a significantly tighter bound than prior methods, due to our usage of $f$-DP. See Fig. 1 for an illustration of these properties, with more details in Section 4.

As a concrete example, consider the release of all state-level US 2020 Census data, which satisfies DP with $\varepsilon = 10.6$ at $\delta = 10^{-10}$. A standard risk analysis (Salem et al., 2023) indicates that the maximum difference between success and baseline probabilities of an attribute inference attack is $> 99$ percentage points (pp), a more involved numeric analysis (Balle et al., 2022) based on the notion of Rényi DP (Mironov, 2017) yields 73pp, whereas our worst-case bound yields 52pp. Moreover, if we assume an adversary with a specific goal and background knowledge, e.g., inferring a disease status with a 1 in 10,000 prevalence in the population, our bound yields $< 0.001$pp. This corroborates and generalizes prior observations (Ziller et al., 2024; Hayes et al., 2024) that even $\varepsilon \approx 10$ attains meaningful privacy guarantees in certain regimes, to arbitrary $f$-DP mechanisms.

To summarize, our contributions are:

- A unifying framework for analyzing re-identification, attribute inference, and data reconstruction risks in the strong threat model of DP. This includes a new formalization of re-identification risk as a variant of the notion of predicate singling out (Cohen and Nissim, 2020), adapted to the standard threat model of DP.
- Demonstration that our bounds offer tighter estimates of risk compared to prior approaches (Cohen and Nissim, 2020; Balle et al., 2022).
- Case studies demonstrating that our results provide a more precise and comprehensive interpretation of privacy risk for the US 2020 Census than prior methods, and enable noise reduction in DP mechanisms by approximately $20\%$, yielding utility improvements in deep learning with DP-SGD of up to 18pp in test accuracy when calibrating noise to a given level of privacy risk (Kulynych et al., 2024).
- Novel bounds on generalization and memorization under $f$-DP as corollaries of our results.
- We release the code as part of the Python package:



https://github.com/Felipe-Gomez/riskcal



Overall, our unifying perspective provides a principled and easy-to-use framework for interpreting and calibrating the degree of privacy protection in DP mechanisms against specific levels of privacy risks that are relevant in practice.

## 2 Background

This section sets up the notation and provides a concise overview of relevant background. We defer a more extensive overview of relevant background to Appendix A.

Suppose that we have a sensitive dataset, e.g., patient healthcare records, $S \in 2^{\mathbb{D}}$, where $\mathbb{D}$ is the data record space. We want to run a randomized algorithm $M(S)$, e.g., a statistical analysis, or train a machine learning model, and release its output. We also call $M(\cdot)$ a *mechanism*. We denote the space of its outputs as $\Theta$, and a specific output, e.g., a model, as $\theta \in \Theta$. We say that two datasets $S, S'$ are *neighbouring* if they belong to a neighborhood relation denoted as $S \simeq S'$. To capture a meaningful notion of training data privacy, this relation must correspond to adding, removing, or replacing all of the data that contains information of a single *privacy unit*, e.g., individual or secret. In the rest of the paper, we follow the standard practice and assume that the privacy unit corresponds to a single record. We consider two standard neighbourhood relations: the *add-remove* relation in which either $S = S' \cup \{z\}$ or $S' = S \cup \{z\}$ for some $z \in \mathbb{D}$, and the *replace-one* relation in which both $S$ and $S'$ have the same size, but differ by one record.

**Differential privacy.** An algorithm $M : 2^{\mathbb{D}} \to \Theta$ satisfies $(\varepsilon, \delta)$-DP if for any measurable $E \subseteq \Theta$ and $S \simeq S'$, we have $\Pr[M(S) \in E] \leq e^{\varepsilon} \Pr[M(S') \in E] + \delta$ (Dwork et al., 2014). We also make use of the notion of $\eta$-total-variation (TV) privacy, which is equivalent to satisfying $(0, \eta)$-DP.

**Strong-adversary membership inference.** DP can be completely characterised via a constraint on the success rate of *strong membership inference attacks* (SMIA), which aim to determine whether a given record was part of the input dataset $S$ based on the output of $M(S)$. Formally, given a sensitive record $z \in \mathbb{D}$, and a partial dataset $\bar{S} \in \mathbb{D}^{n-1}$, let $P = M(\bar{S})$ and $Q = M(\bar{S} \cup \{z\})$. SMIA can be seen as a hypothesis test:

$$H_0 : \theta \sim P, \text{ and } H_1 : \theta \sim Q. \tag{1}$$

For a given decision $\phi : \Theta \to \{0, 1\}$ to reject the null hypothesis, we can quantify the adversary's success by characterizing its error rates (Wasserman and Zhou, 2010; Kairouz et al., 2015; Dong et al., 2022): $\alpha_\phi \triangleq \mathbb{E}_P[\phi], \beta_\phi \triangleq 1 - \mathbb{E}_Q[\phi]$, where $\alpha_\phi$ and $\beta_\phi$ are the false positive rate (FPR) and false negative rate (FNR) respectively. For any desired FPR level $\alpha \in [0, 1]$, the Neyman-Pearson lemma guarantees that there exists an optimal test $\phi^*$ which achieves the lowest possible FNR $\beta$. We can thus characterize the SMIA by a *trade-off curve*, a function which shows the lowest FNR achieved by the most powerful test for any level of FPR $\alpha$: $T(P, Q)(\alpha) \triangleq \inf_{\phi: \Theta \to [0,1]} \{\beta_\phi \mid \alpha_\phi \leq \alpha\}$.

This trade-off curve forms the basis of a more general version of DP called $f$-DP: a mechanism $M(\cdot)$ satisfies $f$-DP if for any $S \simeq S'$ and $\alpha \in [0, 1]$ we have that $T(M(S), M(S'))(\alpha) \geq f(\alpha)$. This formulation is more general than DP: a mechanism $M(\cdot)$ is $(\varepsilon, \delta)$-DP iff it satisfies $f$-DP with:

$$f(\alpha) = \max\{0, 1 - \delta - e^\varepsilon \alpha, \ e^{-\varepsilon}(1 - \delta - \alpha)\}. \tag{2}$$

Notably, $f$-DP is closed under *post-processing*: if $M(\cdot)$ satisfies $f$-DP, then so does $g \circ M$ for any deterministic or randomized mapping $g(\cdot)$. We make use of a lesser known representation of $f$-DP:

**Lemma 2.1.** *An algorithm $M : 2^{\mathbb{D}} \to \Theta$ satisfies $f$-DP iff for any measurable $E \subseteq \Theta$ and $S \simeq S'$:*

$$\Pr[M(S) \in E] \leq 1 - f(\Pr[M(S') \in E]). \tag{3}$$

This form also appears in Kifer et al. (2022), and we provide a self-contained proof in Appendix C. Another property of $f$-DP that is relevant to our work is that it implies $\eta$-TV privacy with $\eta = \max_{\alpha \in [0,1]} (1 - f(\alpha) - \alpha)$ (Kaissis et al., 2024). As a result, even though TV privacy on its own is a weak notion of privacy (Vadhan, 2017), any $f$-DP algorithm implies TV privacy for some $\eta \geq 0$.

## 3 Bounding Operational Privacy Risks

Next, we introduce formalizations of standard notions of risk in privacy-preserving statistics and learning: singling out, attribute inference, and data reconstruction. Our main results enable the analysis of these risks under a unifying framework given by $f$-DP.

### 3.1 Threat Model

We focus on the risk within the *strong threat model* (see, e.g., Balle et al., 2022) in which the adversary has access to the workings of the privacy-preserving algorithm $M(\cdot)$, has access to the partial dataset $\bar{S}$ except for one target record $z$, and has side information about the remaining record $z$ in the form of a prior distribution $z \sim P$. Note that this threat model is slightly different from the standard threat model of SMIA (see Section 2) in that we do not assume that the adversary knows the target record exactly. To quantify the adversary's success rate, we consider expectations over the prior distribution $P$ as well as over the randomness of the algorithm, as we detail next.

Although this might seem like an overly strong model, providing security in this setting is desirable because it represents the worst-case scenario for attacks against privacy of individuals. In particular, as the adversary knows the partial dataset, it captures even the case when the adversary can poison the data (Leemann et al., 2024). By bounding the success of the strong adversary, we also bound the success of attacks against the considered privacy unit under other threat models that are weaker or more realistic for a specific application.

In addition, we discuss existing risk notions which assume that the adversary does *not* have access to the partial dataset, but knows the data distribution $S \sim P^n$. The success is then evaluated over $S \sim P^n$. We call this the *average-dataset threat model*. We summarize the difference between these threat models in Table 1 in Appendix H.

## 3.2 Notions of Risk

**Singling-out risk.** We introduce formalizations of singling-out risk based on the notion of predicate singling out (PSO) (Cohen and Nissim, 2020). For the purpose of our work, we use singling-out risk and re-identification risk interchangeably for brevity, although singling out is only a necessary condition for re-identification, and other approaches to defining re-identification (e.g., also capturing inference) exist (Article 29 Data Protection Working Party, 2014).

**Definition 3.1** (PSO security). *For a given $n > 1$, mechanism $M : \mathbb{D}^n \to \Theta$, data distribution $P$ over $\mathbb{D}$, weight $w \in [0, 1/n]$, and adversary $\mathcal{A}_{n,M,P,w} : \Theta \to \mathbb{Q}_{P,w}$ which outputs a predicate that aims to single out one record in the training dataset from the set of* admissible predicates $\mathbb{Q}_{P,w} \triangleq \{p \mid p : \mathbb{D} \to \{0,1\}, \mathbb{E}_P[p] \leq w\}$, *we define the PSO success rate as follows:*

$$\mathsf{succ}_{PSO}(n, M, P, w; \mathcal{A}) \triangleq \Pr_{\substack{S \sim P^n \\ p \leftarrow \mathcal{A}_{n,M,P,w}(M(S))}} \left[ \sum_{z \in S} p(z) = 1 \right], \tag{4}$$

*and the baseline success as:*

$$\mathsf{base}_{PSO}(n, P, w) \triangleq \sup_{p \in \mathbb{Q}_{P,w}} \Pr_{S \sim P^n} \left[ \sum_{z \in S} p(z) = 1 \right] = \sup_{p \in \mathbb{Q}_{P,w}} n \, \mathbb{E}_P[p] \cdot (1 - \mathbb{E}_P[p])^{n-1}. \tag{5}$$

Observe that PSO security does not assume that the adversary has access to the dataset beyond its distribution $S \sim P^n$, thus assumes the *average-dataset threat model*. As we are interested in the strong-adversary model, we introduce a new notion of PSO security where we assume that the adversary has knowledge of the entire dataset except for one element:

**Definition 3.2** (SPSO security). *For a given $n > 1$, $k \geq 2$, mechanism $M : 2^{\mathbb{D}} \to \Theta$, partial dataset $\bar{S} \in \mathbb{D}^{n-1}$, data distribution $P$ over a* candidate set $\mathbb{W} \subseteq \mathbb{D}$ with $|W| = k$, weight $w \in [0, 1]$, *and adversary $\mathcal{A}_{M,\bar{S},P,w} : \Theta \to \mathbb{Q}_{\bar{S},P,w}$, which outputs a predicate that aims to single out the unknown record in the training dataset from the set of predicates $\mathbb{Q}_{\bar{S},P,w} \subseteq \{p \mid p : \mathbb{D} \to \{0,1\}, \sum_{z' \in \bar{S}} p(z') = 0, \mathbb{E}_P[p] \leq w\}$, we define the strong PSO (SPSO) success rate as follows:*

$$\mathsf{succ}_{SPSO}(M, \bar{S}, P, w; \mathcal{A}) \triangleq \Pr_{\substack{z \sim P \\ p \leftarrow \mathcal{A}_{M,\bar{S},P,w}(M(\bar{S} \cup \{z\}))}} [p(z) = 1], \tag{6}$$

*and the baseline success as:*

$$\mathsf{base}_{SPSO}(\bar{S}, P, w) \triangleq \sup_{p \in \mathbb{Q}_{\bar{S},P,w}} \Pr_{z \sim P} [p(z) = 1] = \sup_{p \in \mathbb{Q}_{\bar{S},P,w}} \mathbb{E}_P[p] \leq w. \tag{7}$$

Unlike PSO, in which the adversary aims to single out any record in an unknown dataset, in SPSO, the adversary aims to find a predicate that matches only one record from a prior set of candidates. We provide additional background on PSO in Appendix A.2.

**Attribute inference and reconstruction attacks.** Following Balle et al. (2022), we formalize attribute inference and data reconstruction attacks using the notion of reconstruction robustness:

**Definition 3.3** (SRR security). *For a given $n \geq 1$, mechanism $M : 2^{\mathbb{D}} \to \Theta$, data distribution $P$ over $\mathbb{D}$, partial dataset $\bar{S} \in \mathbb{D}^{n-1}$, loss function $\ell : \mathbb{D} \times \mathbb{D} \to \mathbb{R}$, threshold $\gamma \in \mathbb{R}$, and reconstruction attack $\mathcal{A}_{M,\bar{S},\mathcal{D}} : \Theta \to \mathbb{D}$, we define the strong reconstruction robustness (SRR) success rate as follows:*

$$\mathsf{succ}_{SRR}(M, \bar{S}, P; \mathcal{A}, \ell, \gamma) \triangleq \Pr_{\substack{z \sim P \\ \hat{z} \leftarrow \mathcal{A}_{M,\bar{S},P}(M(\bar{S} \cup \{z\}))}} [\ell(z, \hat{z}) \leq \gamma], \tag{8}$$

*and the baseline success as* $\mathsf{base}_{SRR}(P; \ell, \gamma) \triangleq \sup_{\hat{z} \in \mathbb{D}} \Pr_{z \sim P} [\ell(z, \hat{z}) \leq \gamma]$.

Robustness against attribute inference attacks can be seen as a special case of SRR:

**Definition 3.4** (SAI security). *For a given $n \geq 1$, mechanism $M : 2^{\mathbb{D}} \to \Theta$, set of attributes $\mathbb{A} = \{1, \ldots, k\}$, mapping from records to attributes $a : \mathbb{D} \to \mathbb{A}$, data distribution $P$ over $\mathbb{D}$, partial dataset $\bar{S} \in \mathbb{D}^{n-1}$, and attribute inference attack $\mathcal{A}_{M,\bar{S},P} : \Theta \to \mathbb{A}$, the strong attribute inference (SAI) success rate is defined as:*

$$\mathsf{succ}_{SAI}(M, \bar{S}, P; \mathcal{A}, a) \triangleq \Pr_{\substack{z \sim P \\ \hat{a} \leftarrow \mathcal{A}_{M,\bar{S},P}(M(\bar{S} \cup \{z\}))}} [a(z) = \hat{a}], \tag{9}$$

*and the baseline success as* $\mathsf{base}_{SAI}(P, a) \triangleq \sup_{\hat{a} \in \mathbb{A}} \Pr_{z \sim P} [a(z) = \hat{a}]$.

We review the prior methods to bound these notions of risk under DP in Appendix A.3.

## 3.3 Main Results: Unifying Bounds

Our core contribution is a unifying relationship between the hypothesis-testing interpretation of DP via $f$-DP and the operational notions of privacy risk in the strong adversary model defined in Section 3.2. We achieve this via a general result that connects the expectation of an attack-specific query function $q(z, \theta)$ over the randomness of $\theta \sim M(S \cup \{z\})$ and $z \sim P$ to the maximum "baseline" expectation of the same function over only $z \sim P$:

**Lemma 3.1.** *Suppose that $M : 2^{\mathbb{D}} \to \Theta$ satisfies $f$-DP w.r.t. either add-remove or replace-one relation. Then, for any bounded function $q : \mathbb{D} \times \Theta \to [0, 1]$, any partial dataset $\bar{S} \in \mathbb{D}^{n-1}$ with $n \geq 1$, and any probability distribution $P$ over $\mathbb{D}$, we have:*

$$\mathbb{E}_{z \sim P} \mathbb{E}_{\theta \sim M(\bar{S} \cup \{z\})} [q(z; \theta)] \leq 1 - f \left( \sup_{\theta \in \Theta} \mathbb{E}_{z \sim P} [q(z; \theta)] \right). \tag{10}$$

Intuitively, the function $q(z; \theta)$ can be seen as indicating the degree of success of some attack against the privacy of the record $z$ given the knowledge of a model $\theta$, on a scale from 0 to 1. We provide a proof of this statement, as well as of all the following results, in Appendix C. Technically, the core steps of the proof involve the usage of the representation of $f$-DP in Lemma 2.1, and an application of Jensen's inequality to push the expectation over $z \sim P$ inside of $f(\alpha)$.

We recover the notions of risk in Section 3.2 by taking relevant instantiations of $q(z; \theta)$, e.g., the indicator of a successful reconstruction event, $q(z; \theta) = \mathbb{1}[\ell(z, \mathcal{A}(\theta)) \leq \gamma]$, in the case of reconstruction attacks. As a result, we get a unifying bound on attack success:

**Theorem 3.1** (Informal)**.** *Suppose that $M(\cdot)$ satisfies $f$-DP w.r.t. either add-remove or replace-one relation. For SPSO, SRR, and SAI, it holds that:*

$$\mathsf{succ}_{[SPSO, SRR, SAI]} \leq 1 - f(\mathsf{base}_{[SPSO, SRR, SAI]}). \tag{11}$$

We provide a precise statement including the missing arguments for each risk notion in Appendix C.

**Normalized success.** Multiple prior works have observed that privacy analyses based on attack success alone can be misleading (Guerra-Balboa et al., 2023; Salem et al., 2023; Cohen et al., 2025). Thus, for all notions of risk, we consider an additional representation of risk in terms of success normalized by the baseline via additive *advantage*: $\mathsf{adv} \triangleq \mathsf{succ} - \mathsf{base}$, where we drop the arguments for brevity. As a consequence of Theorem 3.1, we obtain:

**Theorem 3.2** (Informal)**.** *Suppose that $M(\cdot)$ satisfies $f$-DP w.r.t. either add-remove or replace-one relation. For SPSO, SRR, and SAI, it holds that:*

$$\mathsf{adv}_{[SPSO, SRR, SAI]} \leq 1 - f(\mathsf{base}_{[SPSO, SRR, SAI]}) - \mathsf{base}_{[SPSO, SRR, SAI]}. \tag{12}$$

Theorem 3.2 trivially extends to other ways to normalize attack success, such as $\frac{\mathsf{succ} - \mathsf{base}}{1 - \mathsf{base}}$ (see, e.g., Guo et al., 2023) or $\frac{1 - \mathsf{succ}}{1 - \mathsf{base}}$ (see, e.g., Chatzikokolakis et al., 2023). In the rest of the paper, we use additive advantage to express risk.

**Baseline-independent guarantees.** We can also obtain a bound on advantage that is independent of the baseline by just taking the maximum over all baselines. For this, we use the notion of $\eta$-TV privacy, which any $f$-DP algorithm satisfies for some $\eta$ (see Section 2).

**Theorem 3.3** (Informal)**.** *Suppose that the algorithm $M(\cdot)$ satisfies $\eta$-TV privacy w.r.t. either the add-remove or replace-one relation. Then, the advantage of SPSO, SRR, and SAI is bounded:*

$$\mathsf{adv}_{[SPSO, SRR, SAI]} \leq \eta. \tag{13}$$

This result is useful as (1) it enables to measure and communicate the worst-case risk *across all baselines using a single number,* and (2) we can use standard software tools for analyzing privacy in common algorithms such as DP-SGD (Abadi et al., 2016) by evaluating the DP guarantee for $(\varepsilon = 0, \delta = \eta)$-DP without the need to instantiate the full $f$ curve. We detail on these properties next.

## 3.4 Discussion of the Theoretical Results

**Computing the bounds in practice.** Effective usage of our baseline-specific bounds in Theorems 3.1 and 3.2 requires an analysis of a privacy-preserving algorithm in terms of $f$-DP. Although different variants of DP imply $f$-DP, e.g., via Eq. (2) for a single pair of $(\varepsilon, \delta)$ guarantees, such conversions are loose (Kulynych et al., 2024). For simple mechanisms such as Gaussian or Laplace (Dwork et al., 2014), exact trade-off curves are known (Dong et al., 2022). For more complex algorithms such as DP-SGD (Abadi et al., 2016), there are two ways for obtaining the corresponding curve.

First, we can estimate it using Eq. (2) from the privacy profile of the algorithm (Balle et al., 2018), i.e., the set of all attainable $(\varepsilon, \delta)$-DP pairs. This method has been used for privacy auditing previously (Nasr et al., 2023). Second, it is possible to analyze DP-SGD or other algorithms that are compositions of (subsampled) Gaussian and Laplace mechanisms using a direct method (Kulynych et al., 2024). Both approaches provide tight analyses when using state-of-the-art accounting tools as a backbone, such as the Connect-the-Dots accountant (Doroshenko et al., 2022). In particular, prior work (Nasr et al., 2023) has shown that the upper bounds on privacy leakage obtained with the modern accounting techniques can be nearly reached by empirical attacks in the SMIA threat model.

**Tunability.** Our bound on advantage in Theorem 3.2 enables practitioners to tune the baseline risk. In certain data release scenarios, it might be useful to simulate relevant re-identification or attribute inference attacks, estimate their plausible baseline risk, and consider the risk of the release with respect to such a baseline, i.e., $\mathsf{adv} \leq 1 - f(\mathsf{base}) - \mathsf{base}$. As an example, consider releasing outputs of a mechanism trained on tabular medical data which contains a column corresponding to a patient's HIV status. In this case, we might want to evaluate the risk of an adversary inferring the HIV status under standard threat models considered when releasing medical data (El Emam, 2010) such as "marketer" or "journalist". For instance, the baseline risk of a "marketer" adversary—who does not target any specific individual—could be modeled as guessing based on the prevalence of HIV in the general population. The baseline risk of a "journalist" adversary—who is assumed to be able to obtain side information on the target from public sources—could be modeled as guessing based on a target patient's demographics. We leave the derivation of application-specific procedures and guidelines for modeling baseline risk to future work.

In high-risk scenarios, e.g., public model releases, it might be more desirable to ensure security against the worst-case attacks. In this case, it is more appropriate to use the baseline-independent bound of $\mathsf{adv} \leq \max_{\mathsf{base} \in [0,1]} (1 - f(\mathsf{base}) - \mathsf{base}) \leq \eta$ from Theorem 3.3.

**On tightness of the bounds.** Our bounds are never vacuous, and are saturated for perfectly private and blatantly non-private mechanisms. However, they can be significantly tightened under additional assumptions on the adversary's threat model. In particular, we show an example of such strengthening in Appendix D for an important case of *binary attribute inference* with a non-uniform prior, and empirically show that it provides tighter bounds in this setting than prior work (Guo et al., 2023). Although there exist cases under which the unified bound can be tightened, we conjecture that it is tight in the most general case, i.e., there exist non-trivial settings under which Theorem 3.1 holds with equality. We leave the verification of this conjecture, and the identification of settings under which equality is achieved in case it is true, to future work.

**Beyond privacy.** Our results extend to statistical learning theory. Concretely, in Appendix E, we show that our derivations can be extended to a new kind of generalization bound. Informally, we show that for the on-average train set error $\mathsf{err}_{\mathrm{tr}}$ and on-average generalization error $\mathsf{err}_{\mathrm{test}}$ (Shalev-Shwartz et al., 2010) of a learning algorithm satisfying $f$-DP, it holds that $\mathsf{err}_{\mathrm{test}} \leq 1 - f(\mathsf{err}_{\mathrm{tr}})$. Moreover, in Appendix F, we show that our derivations imply a bound on memorization (Feldman, 2019; Zhang et al., 2023). We believe these results are of independent interest.

## 4 Experimental Evaluation

### 4.1 Bounds Comparison

**Singling-out risk.** We compare prior bounds on singling-out risk in the average-dataset threat model from Cohen and Nissim (2020) (detailed in Appendix B) to our bound on SPSO in Theorem 3.2, which operates under a different threat model. To provide an apples-to-apples comparison, we fix

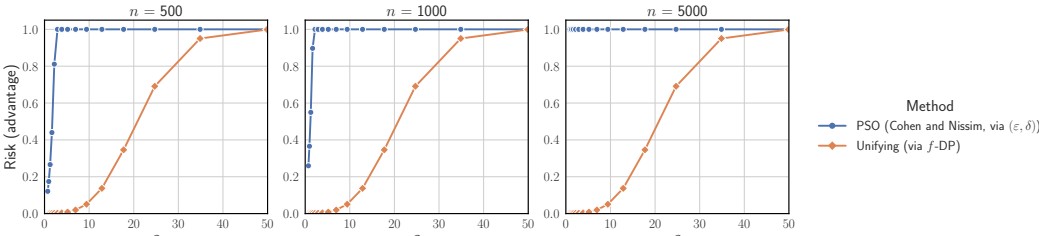

Figure 2: **Our bound on predicate singling out in the strong threat model (SPSO) is always non-vacuous, and, surprisingly, shows significantly lower risk than bounds in the PSO threat model.** The risk is $\mathsf{adv} = \mathsf{succ} - \mathsf{base}$ with fixed given base for Gaussian mechanism with $\varepsilon$ calculated for $\delta = 10^{-5}$.

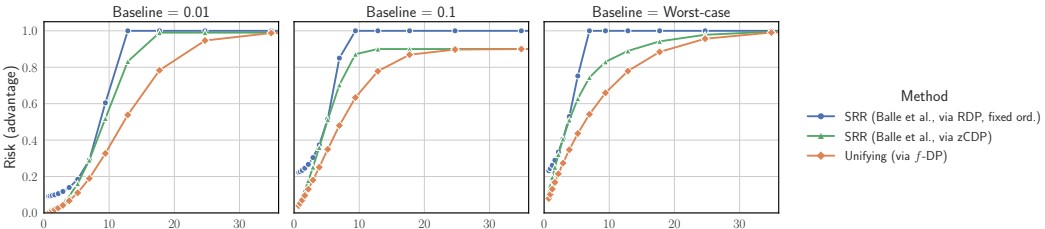

Figure 3: **Our bound on reconstruction robustness shows lower risk than prior bounds.** We show risk as $\mathsf{adv} = \mathsf{succ} - \mathsf{base}$ for three different baseline values for Gaussian mechanism with $\varepsilon$ calculated for $\delta = 10^{-5}$.

the weight $w$ for both threat models. The baseline probability for PSO is $\mathsf{base} = n \cdot w \cdot (1 - w)^{n-1}$, whereas it is $\mathsf{base} = w$ for SPSO. We use standard Laplace and Gaussian mechanisms (Dwork et al., 2014) with varying noise levels, and simulate different dataset sizes $n \in \{500, 1000, 5000\}$ with a fixed weight $w = 1/5000$. We use $\delta = 10^{-5}$ to derive $\varepsilon$ for the Gaussian mechanism, and analyze DP under replace-one relation.

We show the results for the Gaussian mechanism in Fig. 2 (the results for Laplace mechanism are similar, shown in Appendix H). Unlike the PSO bounds, our bounds provide meaningful guarantees even in the $\varepsilon \in [10, 20]$ regime, and saturate only around $\varepsilon \approx 35$. Note that our SPSO bound based on $f$-DP and the strong adversary threat model shows *lower* risk than the PSO bound in the average-dataset threat model based on $(\varepsilon, \delta)$-DP. In Appendix F, we derive a novel PSO guarantee based on $f$-DP, but we observe no meaningful difference from the $(\varepsilon, \delta)$-DP bound in Fig. 2 (see Appendix H). We hypothesize that both phenomena are due to the looseness in the derivations of the average-dataset PSO results that are not present in the SPSO derivation. We leave the derivation of tighter PSO bounds as future work.

**Reconstruction robustness.** Next, we compare prior bounds on SRR from Balle et al. (2022) that are based on Rényi DP (RDP) (Mironov, 2017) and zero-concentrated DP (zCDP) (Dwork and Rothblum, 2016; Bun and Steinke, 2016), detailed in Appendix A.3, to our unifying bounds in Theorem 3.2. We use the Gaussian mechanism, analyzed under the add-one relation. We vary the noise parameter $\sigma$, and compute $\varepsilon$ at $\delta = 10^{-5}$. For the SRR bounds, we evaluate separately the bound for a single RDP guarantee $(t, \varepsilon)$ with $t = 2$, and the bound for the tighter zCDP guarantee. Fig. 3 demonstrates that our bound always shows lower risk than prior approaches.

## 4.2 Case Studies

We present two case studies which showcase how our results can be used to interpret and calibrate realistic DP mechanisms. Additionally, in Appendix G, we discuss another application of our results to a DP statistical-query answering system (Gaboardi et al., 2020).

**Calibrating noise to reconstruction risk in deep learning.** It is possible to partially reconstruct training-data records from observing outputs or weights of classification models (Balle et al., 2022) and language models (Carlini et al., 2022). In this case study, we assume the modeler fine-tunes a

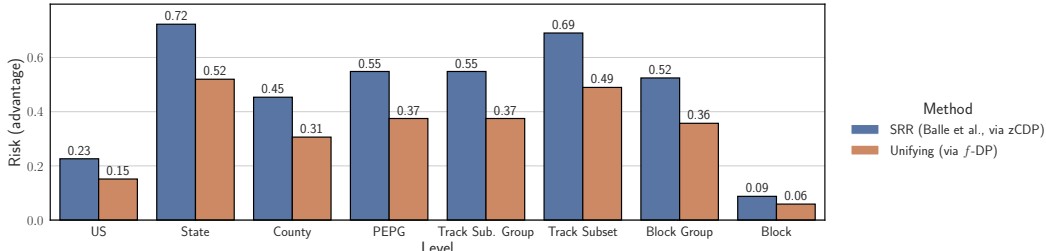

Figure 4: **Our method shows up to 33% lower worst-case reconstruction risk in the US 2020 Census release than the prior method.** x axis shows granularity levels of the release, y axis shows risk of attacks as $\mathsf{succ} - \mathsf{base}$ for the worst-case baseline.

language model for text sentiment classification on the SST-2 dataset (Socher et al., 2013). In order to limit the privacy risk to a given threshold, e.g., ensure at most $0.15$ increase in the probability of successful reconstruction under the worst-case baseline, the modeler runs DP-SGD with attack-aware noise calibration (Kulynych et al., 2024). To simulate this, we fine-tune multiple GPT-2 (small) models (Radford et al., 2019) using a DP version of LoRA (Yu et al., 2021) (we provide technical details in Appendix G). We obtain the $f$ curve under the add-remove relation for each model using the direct method (Kulynych et al., 2024), and apply Theorem 3.2 to measure risk. We compare this to the RDP-based analysis from Balle et al. (2022), where we optimize RDP over a grid (see Appendix A.3).

Fig. 1 shows that if we calibrate the noise scale to a given level of maximum attack risk, our unifying bound enables one to choose a lower noise scale at the same level of risk, which, in turn, results in better classification accuracy. Fig. 1 (left) shows the risk as a function of noise scale, and Fig. 1 (middle) shows the corresponding accuracy when training using that noise scale. The figure demonstrates that for the worst-case risk target of $0.15$, we can increase the accuracy from $52\%$ to $70\%$ using our analysis, as opposed to an RDP-based one. Notably, the increase in accuracy is due to a more precise privacy calculation alone, thus does not come with additional costs. In Appendix F, we detail on a variant of this case study with CIFAR-10 (Krizhevsky et al., 2009).

**Analyzing risk in US Census release.** The US Census bureau has released the 2020 Census data using DP (Abowd et al., 2022). In this case study, we aim to analyze this release in terms of operational privacy risks using our bounds, leveraging a recent analysis of the Census algorithm in terms of $f$-DP (Su et al., 2024). The Census data consists of eight different levels of granularity, ranging from the US-wide level to block-level, with different privacy guarantees at each level. The standard analysis uses zCDP, thus we compare Theorem 3.2 to the analysis of data reconstruction risk via zCDP, as in our previous experiments in Section 4.1.

In Fig. 4, we present the results for each granularity level under the worst-case baseline, which show that our unifying bound indicates up to $33\%$ less risk. The results for other baselines are similar, shown in Fig. 9, Appendix H. Moreover, in Fig. 1 (right) we show the risk as a function of baseline for the state-level release as an illustration for analyzing risk at different baselines.

# 5   Related Work

Prior work has extensively studied bounds on attack risks in DP. Balle et al. (2022) introduced a formalization of robustness to reconstruction attacks and provided bounds using $(\varepsilon, 0)$-DP, RDP and zCDP. Compared to these, our bounds are tighter, as we demonstrate in Section 4. Under specialized threat models, these bounds were subsequently improved by Guo et al. (2023), and by Guerra-Balboa et al. (2023) for the case of $(\varepsilon, 0)$-DP. We study the bounds by Guo et al. (2023), which specifically assume a discrete prior $P$, in Appendix D. Cherubin et al. (2024) provided closed-form bounds on data reconstruction and membership inference advantage for DP-SGD using an approximate analysis. Our results do not rely on approximations and instead use the tight privacy analysis in terms of $f$-DP based on the state-of-the-art DP-SGD accountants, as detailed in Section 3.4.

Hayes et al. (2024) used a decision-theoretic view of DP via approximations to provide a bound on reconstruction robustness. There are several core differences compared to our results: (1) they provide a Monte-Carlo sampling-based method rather than a provable bound, (2) their method is

specific to DP-SGD, and (3) it only covers reconstruction attacks. In contrast, our work focuses on the fundamental connection between different standard types of risks and the $f$-DP framework, as opposed to providing a method to evaluate a specific type of privacy risk in DP-SGD. Our approach enables us to obtain provable bounds on risks for general DP mechanisms. Even though the theoretical underpinnings of our approaches are different, the outputs of the algorithm from Hayes et al. (2024) will converge in the limit of infinite Monte-Carlo samples to our $f$-DP bound in the setting of reconstruction risk in DP-SGD.

In Appendix B, we provide an additional overview of the alternative Bayesian approaches to interpret DP and other approaches to unify privacy risks.

**Bibliographic note.** This paper subsumes the results from a prior workshop paper (Kaissis et al., 2023a) presented at the 2023 Theory and Practice of Differential Privacy (TPDP) workshop.

## 6 Concluding Remarks

This paper presents a unifying framework for analyzing re-identification, attribute inference, and data reconstruction risks—realistic privacy threats facing the releases of statistics, models, or de-identified data—using $f$-differential privacy, a decision-theoretic approach to differential privacy. We demonstrated that our unifying framework is applicable to privacy-preserving machine-learning algorithms such as DP-SGD, and statistical applications such as the US Census data release.

**Impact.** We have shown that the existing approaches to mapping DP parameters to risk are imprecise. As a result, in settings where ensuring a certain maximum level of operational risk is required—as opposed to ensuring a target value of $\varepsilon$—practitioners would need to add more noise than is necessary. Not only does this hurt overall utility, but has negative effects on fairness (Bagdasaryan et al., 2019) and prediction consistency (Kulynych et al., 2023). Our approach provides a more precise way to estimate maximum risk, enabling practitioners to obtain better utility at the same level of target risk.

**$f$-DP for analyzing operational risks.** Our empirical and theoretical results demonstrated that the decision-theoretic view of DP via the $f$-DP framework is a useful tool for analyzing privacy-preserving algorithms in terms of operational risks. As we have shown, it enables both substantially tighter characterizations of operational risk compared to other standard approaches such as Rényi DP or Concentrated DP, and easy-to-use unified analyses of various types of risk.

**Optimality.** Although our bounds outperform prior methods to measure risk in DP, as we show in Appendix D, they can be significantly strengthened under additional assumptions on the adversarial model, e.g., by assuming the setting of binary attribute inference. We leave finding other practically relevant settings in which the bounds can be further tightened—either under specific attack configurations and priors like in Appendix D or under relaxed threat models (see, e.g., Kaissis et al., 2023b; Swanberg et al., 2025)—to future work.

**Importance of the privacy unit.** Our privacy risk bounds have to be interpreted in terms of the underlying privacy unit. For instance, when training language models on open internet data, it is often practically infeasible to define a neighborhood relation that protects the privacy of *individuals*, as an individual's sensitive data can appear in various forms across multiple documents (Brown et al., 2022). In such settings, a more feasible privacy unit is a single *document* (Sinha et al., 2025). Thus, our bounds in this context would quantify the risk to a single *document* rather than to an *individual*.

**The need for transparent processes to prevent baseline gaming.** Our methods provide a way to adjust the level of risk to a given adversarial setting by modeling the baseline risk. This is a strength as it enables to measure the risk of relevant threats, but also could facilitate privacy theater (Smart et al., 2022) if the communicated or calibrated risk is specific to a baseline that is too low. In this case, the risk estimate could appear lower than it actually is. Trustworthy usage of our baseline-specific bounds requires ensuring that the baseline is well-justified and representative of realistic threats.

## Acknowledgments

The authors would like to thank Thanh-Long Tran for spotting the missing $\Lambda(\theta) = t$ case in the original version of the proof of Lemma 2.1 in Appendix C. This project is supported by the U.S. Department of Energy, Office of Science, Office of Advanced Scientific Computing Research, Department of Energy Computational Science Graduate Fellowship under Award Number DE-SC0022158, by Swiss National Science Foundation under Award Numbers 10003518 and 237378, and is part of the SYNTHIA project. SYNTHIA (Synthetic Data Generation framework for integrated validation of use cases and AI healthcare applications) is supported by the Innovative Health Initiative Joint Undertaking (IHI JU) under grant agreement No. 101172872. Thus, the project is partially funded by the European Union, the private members, and those contributing partners of the IHI JU. Views and opinions expressed are however those of the authors only and do not necessarily reflect those of the aforementioned parties. Neither of the aforementioned parties can be held responsible for them.

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

# A Extended Background

This section presents a more extensive overview of the background.

## A.1 Differential Privacy and its Variants

**Classical differential privacy.** Differential privacy (DP) is a formal notion of privacy applicable to releases of statistical information or machine learning models:

**Definition A.1** (Dwork et al., 2006). *A mechanism $M : 2^{\mathbb{D}} \to \Theta$ satisfies $(\varepsilon, \delta)$-DP if for any $S \simeq S'$ we have $\mathsf{E}_{e^\varepsilon}(P \parallel Q) \leq \delta$, where $P$ and $Q$ are the respective probability distributions of $M(S)$ and $M(S')$, and the* hockey-stick divergence *between probability distributions is defined as follows:*

$$\mathsf{E}_\gamma(P \parallel Q) \triangleq \sup_{E \subseteq \Theta} P(E) - \gamma Q(E). \tag{14}$$

*We refer to the case when $\delta = 0$ as* pure DP.

Realistic DP mechanisms satisfy a potentially infinite collection of different $(\varepsilon, \delta)$-DP guarantees. To analyze this collection, we make use of a notion of the privacy profile:

**Definition A.2** (Balle et al., 2018). *A mechanism $M(\cdot)$ has a privacy profile $\delta(\varepsilon)$ if for every $\varepsilon \in \mathbb{R}$, it satisfies $(\varepsilon, \delta(\varepsilon))$-DP.*

**Strong-adversary membership inference.** DP can be completely characterised via a constraint on the success rate of *membership inference attacks*, which aim to determine whether a given record was part of the input dataset $S$ based on the output of $M(S)$. Formally, given a sensitive record $z \in \mathbb{D}$, and a partial dataset $\bar{S} \in \mathbb{D}^{n-1}$, consider a probability distribution of the algorithm's outputs when the record is not part of the dataset, $P(E) \triangleq \Pr_{\theta \sim M(\bar{S})}[\theta \in E]$, and the respective distribution when the record is present in the data, $Q(E) \triangleq \Pr_{\theta \sim M(\bar{S} \cup \{z\})}[\theta \in E]$. In a *strong-adversary membership inference attack* (see, e.g., Nasr et al., 2021), given knowledge of the mechanism $M(\cdot)$, its output $\theta$, a target record $z$, and a partial dataset $\bar{S}$, the adversary aims to determine whether the record was part of the training dataset used to produce $\theta$ or not. In other words, the attack is a hypothesis test which aims to distinguish between two hypotheses:

$$H_0 : \theta \sim P, \text{ and } H_1 : \theta \sim Q. \tag{15}$$

"Strong" refers to the fact that the adversary knows all of the partial dataset $\bar{S}$. We discuss this threat model further in Section 3.1.

**Differential privacy as hypothesis testing.** For a given hypothesis test (equivalent to a membership inference attack) $\phi : \Theta \to \{0, 1\}$ which returns 0 when the guess is $H_0$, and 1 when the guess is $H_1$, we can quantify the adversary's success by characterizing the error rates of the test (Wasserman and Zhou, 2010; Kairouz et al., 2015; Dong et al., 2022):

$$\alpha_\phi \triangleq \mathbb{E}_P[\phi], \quad \beta_\phi \triangleq 1 - \mathbb{E}_Q[\phi], \tag{16}$$

where $\alpha_\phi$ and $\beta_\phi$ are the false positive rate (FPR) and false negative rate (FNR) respectively.

For any desired FPR level $\alpha \in [0, 1]$, the Neyman-Pearson lemma guarantees that there exists an optimal test $\phi^*$ which achieves the lowest possible FNR $\beta$. We can thus characterize the attack setting by a *trade-off curve*, a function which shows the lowest FNR achieved by the most powerful level $\alpha$ test for any level of FPR $\alpha \in [0, 1]$:

$$T(P, Q)(\alpha) \triangleq \inf_{\phi: \Theta \to [0,1]} \{\beta_\phi \mid \alpha_\phi \leq \alpha\} \tag{17}$$

A common way to summarize the success of the optimal strong-adversary membership inference attack (SMIA) is via its *advantage*, the difference between its TPR and FPR (Yeom et al., 2018):

$$\mathsf{adv}_{\text{SMIA}} \triangleq \sup_{\phi: \Theta \to [0,1]} 1 - \beta_\phi - \alpha_\phi.$$

We can now define $f$-DP, a version of DP which centers the hypothesis-testing interpretation:

**Definition A.3** (Dong et al., 2022). *A mechanism $M(\cdot)$ satisfies $f$-Differential Privacy ($f$-DP) if, for any pair $S \simeq S'$, the trade-off curve of the strong membership inference test is bounded:*

$$T(M(S), M(S'))(\alpha) \geq f(\alpha), \quad \forall \alpha \in [0, 1], \tag{18}$$

*where $f : [0, 1] \to [0, 1]$ is a* valid trade-off function: *a convex, continuous, non-increasing function such that $f(\alpha) \leq 1 - \alpha$ for all $\alpha \in [0, 1]$.*

Note that $(\varepsilon, \delta)$-DP is a special case of $f$-DP. Specifically, a mechanism is $(\varepsilon, \delta)$-DP if and only if it satisfies $f$-DP with the following $f(\alpha)$:

$$f(\alpha) = \max\{0, 1 - \delta - e^\varepsilon \alpha, \ e^{-\varepsilon}(1 - \delta - \alpha)\}. \tag{19}$$

We make use of the following equivalent representation of $f$-DP:

**Lemma 2.1.** *An algorithm $M : 2^{\mathbb{D}} \to \Theta$ satisfies $f$-DP iff for any measurable $E \subseteq \Theta$ and $S \simeq S'$:*

$$\Pr[M(S) \in E] \leq 1 - f(\Pr[M(S') \in E]). \tag{3}$$

This form is key to deriving our results in a streamlined way.

A useful property of $f$-DP is *post-processing*: if $M(\cdot)$ satisfies $f$-DP, so does $g \circ M$ for any deterministic or randomized mapping $g(\cdot)$. In other words, post-processing the results of an $f$-DP mechanism cannot decrease its privacy or, equivalently, cannot increase the success of the aforementioned decision problem. This is a consequence of the Blackwell-Sherman-Stein theorem (Dong et al., 2022; Kaissis et al., 2024; Su, 2024).

**Total-variation privacy.**   We also make use of the notion of total variation (TV) privacy:

**Definition A.4.** *A mechanism $M : 2^{\mathbb{D}} \to \Theta$ satisfies $\eta$-TV privacy if it satisfies $(0, \eta)$-DP. Equivalently, the total variation distance, a special case of $\mathsf{E}_\gamma$ with $\gamma = 1$, is bounded for all $S \simeq S'$:*

$$\mathsf{E}_1(P \parallel Q) = \sup_{E \subseteq \Theta} P(E) - Q(E) \leq \eta, \tag{20}$$

*where $P$ and $Q$ are the respective probability distributions of $M(S)$ and $M(S')$.*

TV privacy has been extensively studied under equivalent forms and different names in literature (Geng and Viswanath, 2015; Bassily et al., 2016; Kulynych et al., 2022a; Chatzikokolakis et al., 2023; Ghazi and Issa, 2023). Although TV privacy is a weak notion of privacy on its own (Vadhan, 2017), any DP mechanism satisfies TV privacy for some $\eta$:

**Proposition A.1.** *The following statements hold:*

- *$(\varepsilon, \delta)$-DP implies $\eta$-TV privacy with $\eta = \frac{e^\varepsilon - 1 + 2\delta}{e^\varepsilon + 1}$ (Kairouz et al., 2015).*
- *$f$-DP implies $\eta$-TV privacy with $\eta = \max_{\alpha \in [0,1]}(1 - f(\alpha) - \alpha)$ (Kaissis et al., 2024).*

**Privacy via Rényi divergence.**   Another notion we use is Rényi DP (RDP) (Mironov, 2017).

**Definition A.5** (Mironov, 2017). *A mechanism $M(\cdot)$ satisfies $(t, \varepsilon)$-RDP iff for all $S \simeq S'$:*

$$D_t(P \parallel Q) \leq \varepsilon, \tag{21}$$

*where $P$ and $Q$ are the respective probability distributions of $M(S)$ and $M(S')$, and $D_t(P \parallel Q)$ is the Rényi divergence of order $t \geq 1$:*

$$D_t(P \parallel Q) \triangleq \frac{1}{t - 1} \log \mathbb{E}_{o \sim Q} \left( \frac{P(o)}{Q(o)} \right)^t,$$

*where the case of $t = 1$ is defined by continuous extension.*

The mechanism satisfies $\rho$-zCDP (Bun and Steinke, 2016; Dwork and Rothblum, 2016) for $\rho \geq 0$ if it satisfies $(t, \rho\, t)$-RDP for every $t \geq 1$.

## A.2 Predicate Singling Out

In this section, we provide a brief overview of the notion of predicate singling out security.

Predicate singling out (PSO) risk, which is defined in Section 3.2, is a privacy concept introduced by Cohen and Nissim (2020) to rigorously interpret the "singling out" criterion mentioned in the EU GDPR (General Data Protection Regulation). Their goal was to bridge the gap between legal expectations of data anonymity and technical guarantees provided by data anonymization mechanisms. At a high level, in the PSO threat model, the adversary uses the output of a data release mechanism $\theta = M(S)$ to find a predicate $p$ that matches exactly one individual (row) in the original dataset $S$ with significantly greater success than would be expected by chance. In this context, a predicate represents a set of attribute values that characterize a person, e.g., "speaks Dutch $\wedge$ vegan $\wedge$ concert pianist $\wedge$ born March 15." If a predicate matches only one individual in a dataset, that individual is singled out. Moreover, such a predicate is unlikely to identify a person by random chance. Formally, the goal of the adversary is to find a predicate $p$ such that $\sum_{z \in S} p(z) = 1$.

Unlike differential privacy, which operates under a strong threat model to provide formal "upper bounds" to more realistic notions of privacy, PSO takes the alternate approach of intentionally being a necessary but not sufficient condition for privacy, i.e. a "lower bound" to realistic notions of privacy. The goal of the original paper was to make claims such as "if you do not satisfy PSO security, you are not GDPR compliant". Hence, Cohen and Nissim consider the i.i.d. setting: the dataset $S$ is sampled i.i.d. from a distribution $P$ known to the adversary. Cohen and Nissim reason about this adversary by considering the baseline (before seeing $M(S)$) and success probability (after seeing $M(S)$) of the adversary assuming the adversary's predicate in both cases has a fixed *weight* $w$, where weight is defined as $w = \mathbb{E}_P[p]$, the probability that the predicate $p$ evaluates to 1 on a random sample from $P$.

In this setting, the optimal baseline is the random guessing strategy: before observing the output of the mechanism $M(S)$, the adversary constructs a predicate $p$ that has a small weight $w$. Intuitively, predicates such as "woman $\wedge$ over 60 year old" have high weight and predicates such as the Dutch vegan pianist from the previous paragraph have low weight. The best the adversary can do as a baseline is to pick a predicate with weight $w$ and hope that they single out. This has a probability of $\mathsf{base} = nw(1-w)^{n-1}$ for a dataset of size $n$.

## A.3 Previous Bounds on Privacy Risk Notions

**PSO Security**    Assuming a fixed weight $w$ and approximate DP, Cohen and Nissim (2020) showed that the optimal strategy for PSO can be upper bounded as a function of the baseline probability:

**Theorem A.1** (Cohen and Nissim, 2020). *Suppose that $M : \mathbb{D}^n \to \Theta$ satisfies $(\varepsilon, \delta)$-DP w.r.t. replace-one neighbourhood relation. Then, for any given $n > 1$, data distribution $P$ over $\mathbb{D}$, and an adversary $\mathcal{A}_{n,M,P,w} : \Theta \to \mathbb{Q}_P$ with $\mathbb{Q}_P \triangleq \{p \mid p : \mathbb{D} \to \{0,1\}, \mathbb{E}_P[p] \le w\}$ for given $w \in [0, {}^1/n]$, we have:*

$$\mathsf{succ}_{PSO}(n, M, P, w; \mathcal{A}) \le n(e^\varepsilon w + \delta) \tag{22}$$

$$= e^\varepsilon \cdot \frac{\mathsf{base}_{PSO}(n, P, w)}{(1-w)^{n-1}} + n\delta. \tag{23}$$

This is the bound we use in Fig. 2.

**Reconstruction Robustness**    Next, we review the state-of-the-art results for bounding reconstruction robustness. We start with the Rényi DP bound due to Balle et al. (2022):

**Theorem A.2** (Balle et al., 2022). *Suppose that $M(\cdot)$ satisfies $(t, \varepsilon)$-RDP w.r.t. either add-remove or replace-one relation. It holds that:*

$$\mathsf{succ}_{[SRR]} \le \left(\mathsf{base}_{[SRR]} \cdot e^\varepsilon\right)^{\frac{t-1}{t}}. \tag{24}$$

Applying the above result to a $\rho$-zCDP mechanism and optimizing $t$ to minimize the upper bound yields the following:

**Corollary A.1** (Balle et al., 2022). *Suppose that $M(\cdot)$ satisfies $\rho$-zCDP w.r.t. either add-remove or replace-one relation. It holds that:*

$$\mathsf{succ}_{[SRR]} \leq \exp\left\{ -\left( \sqrt{\log \frac{1}{\mathsf{base}_{[SRR]}}} - \sqrt{\rho} \right)^2 \right\}. \tag{25}$$

Corollary A.1 is the bound on *SRR* used in Figs. 3, 4, 5 and 9. We can also apply Theorem A.2 to a mechanism that satisfies a continuum of RDP guarantees and minimize over the upper bound:

**Corollary A.2.** *Suppose that $M(\cdot)$ satisfies $(t, \varepsilon(t))$-RDP for all $t > 1$ w.r.t. either add-remove or replace-one relation. It holds that:*

$$\mathsf{succ}_{[SRR]} \leq \min_{t > 1} \left\{ \left(\mathsf{base}_{[SRR]} \cdot e^{\varepsilon(t)}\right)^{\frac{t-1}{t}} \right\}. \tag{26}$$

We use Corollary A.2 in Fig. 1 as the prior method that we compare our $f$-DP bound to.

# B  Additional Discussion on Related Work

**Related unification efforts.**  Recently, Cohen et al. (2025) proposed a unifying notion of risk called *narcissus resiliency*, which averages over dataset sampling. The notions of risk we consider under our unifying framework are all tailored to the strong threat model, which is dataset-specific. We additionally provide bounds on Narcissus resiliency using $f$-DP in Appendix F. Salem et al. (2023); Cummings et al. (2024) provided taxonomies and reductions between risk notions. Our focus is providing analysis using $f$-DP, as opposed to a taxonomization.

**Bayesian semantics of DP.**  An alternative way to analyze and interpret risk in DP is a Bayesian posterior-to-prior analysis (see, e.g., Wood et al., 2018; Kifer et al., 2022; Kazan and Reiter, 2024). We focus on security-based literature instead, where risk is analyzed from the point of view of relevant adversarial models. See Kifer et al. (2022); Kaissis et al. (2024) for connections between the Bayesian view and $f$-DP.

**Attack-aware noise calibration.**  Kulynych et al. (2024) introduced attack-aware noise calibration, the idea of calibrating DP algorithms to a given level of operational attack risk as opposed to the standard practice of calibrating to a given pair of $(\varepsilon, \delta)$ parameters. They evaluated this approach for the use case of calibrating noise to a given level of membership inference risk. One of the applications of our result is that it extends the toolbox of attack-aware calibration to further attack risks, thus enabling practitioners to calibrate DP algorithms to notions of risk beyond membership inference, which often appear in data protection guidelines (see Section 1).

# C  Omitted Proofs

## C.1  Additional and Auxiliary Results

We start with a proof of a useful equivalent form of $f$-DP.

**Lemma 2.1.** *An algorithm $M : 2^{\mathbb{D}} \to \Theta$ satisfies $f$-DP iff for any measurable $E \subseteq \Theta$ and $S \simeq S'$:*

$$\Pr[M(S) \in E] \leq 1 - f(\Pr[M(S') \in E]). \tag{3}$$

*Proof.* Note that we only consider the functions $f$ that are valid trade-off functions (Definition A.3). Denote by $P$ the distribution of $M(S)$ and by $Q$ the distribution of $M(S')$.

$\implies$ Suppose that the algorithm satisfies $f$-DP. Let $E \subseteq \Theta$ be any measurable set. Consider the deterministic test $\phi(\theta) = \mathbb{1}[\theta \in E]$. By the definition of $f$-DP, for this specific test we must have:

$$f(\mathbb{E}_Q[\phi]) \leq 1 - \mathbb{E}_P[\phi].$$

Substituting $\mathbb{E}_P[\phi] = P(E)$ and $\mathbb{E}_Q[\phi] = Q(E)$, we directly obtain:

$$f(Q(E)) \leq 1 - P(E).$$

$\Longleftarrow$ Suppose that Eq. (3) holds for any measurable $E \subseteq \Theta$ and any $S \simeq S'$. Let $\phi : \Theta \to [0,1]$ be any hypothesis test. By the Neyman-Pearson lemma, it suffices to consider likelihood-ratio tests. Any such test can be decomposed into a convex combination of two indicator functions:

$$\phi(\theta) = \mathbf{1}[\Lambda(\theta) > t] + c \cdot \mathbf{1}[\Lambda(\theta) = t]$$
$$= (1 - c) \cdot \mathbf{1}[\Lambda(\theta) > t] + c \cdot \mathbf{1}[\Lambda(\theta) \geq t],$$

for some $c \in [0,1]$ and $t \in \mathbb{R}$, where $\Lambda(\theta) \triangleq \mathrm{d}P/\mathrm{d}Q(\theta)$ is the likelihood ratio.

Denoting by $E_1 = \{\theta \in \Theta \mid \Lambda(\theta) > t\}$ and $E_2 = \{\theta \in \Theta \mid \Lambda(\theta) \geq t\}$, we have by the linearity of expectation and the convexity of $f$:

$$f(\mathbb{E}_Q[\phi]) = f((1 - c) \cdot Q(E_1) + c \cdot Q(E_2))$$
$$\leq (1 - c) \cdot f(Q(E_1)) + c \cdot f(Q(E_2)).$$

By assumption that Eq. (3) holds for the sets $E_1$ and $E_2$:

$$\leq (1 - c) \cdot (1 - P(E_1)) + c \cdot (1 - P(E_2))$$
$$= 1 - \mathbb{E}_P[\phi].$$

As $\alpha_\phi = \mathbb{E}_Q[\phi]$ and $\beta_\phi = 1 - \mathbb{E}_P[\phi]$, this implies $f(\alpha_\phi) \leq \beta_\phi$, which recovers Eq. (18). $\qquad\square$

To show the main result on the unified bounds, we make use of the following elementary lemmas. First, due to the properties of the trade-off function $f$, we can push the expectation inside of $f(\cdot)$:

**Lemma C.1.** *Suppose that $f$ is a valid trade-off function according to Definition A.3. For any random variable $W$ over $[0,1]$, we have:*

$$\mathbb{E}[1 - f(W)] \leq 1 - f(\mathbb{E}\,W). \tag{27}$$

*Proof.* By assumption, $1 - f$ is concave and increasing. The result immediately follows from applying the linearity of expectation and Jensen's inequality. $\qquad\square$

Second, we can supremize an expectation inside of $f(\cdot)$:

**Lemma C.2.** *Suppose that $f$ is a valid trade-off function according to Definition A.3. For any random variable $V$ taking values in a set $\mathbb{V}$, and any bounded function $g : \mathbb{V} \to [0,1]$, we have:*

$$1 - f(\mathbb{E}\,g(V)) \leq 1 - f\left(\sup_{v \in \mathbb{V}} g(v)\right). \tag{28}$$

*Proof.* Observe that $\mathbb{E}\,g(V) \leq \sup_{v \in \mathbb{V}} g(v)$. We get the result as $1 - f$ is increasing. $\qquad\square$

Next, we can show a result which can be seen as a version of a risk bound for on-average baselines.

**Lemma C.3.** *Suppose that $M : 2^{\mathbb{D}} \to \Theta$ satisfies $f$-DP w.r.t add-remove relation. Then, for any bounded function $q : \mathbb{D} \times \Theta \to [0,1]$, any partial dataset $\bar{S} \in \mathbb{D}^n$, and any probability distribution $P$ over $\mathbb{D}$, we have:*

$$\mathbb{E}_{z \sim P}\,\mathbb{E}_{\theta \sim M(\bar{S} \cup \{z\})}\,[q(z; \theta)] \leq 1 - f\left(\mathbb{E}_{z \sim P}\,\mathbb{E}_{\theta \sim M(\bar{S})}\,[q(z; \theta)]\right). \tag{29}$$

*Moreover, if the $M(\cdot)$ satisfies $f$-DP w.r.t. replace-one relation, we have for any $z' \in \mathbb{D}$:*

$$\mathbb{E}_{z \sim P}\,\mathbb{E}_{\theta \sim M(\bar{S} \cup \{z\})}\,[q(z; \theta)] \leq 1 - f\left(\mathbb{E}_{z \sim P}\,\mathbb{E}_{\theta \sim M(\bar{S} \cup \{z'\})}\,[q(z; \theta)]\right). \tag{30}$$

*Proof.* Fix any $z \in \mathbb{D}$. By the form of $f$-DP in Lemma 2.1 and the post-processing property of $f$-DP, we have:

$$\mathbb{E}_{\theta \sim M(\bar{S})}\,q(z; \theta) \leq 1 - f\left(\mathbb{E}_{\theta \sim M(\bar{S} \cup \{z\})}\,q(z; \theta)\right)$$
$$\mathbb{E}_{\theta \sim M(\bar{S} \cup \{z\})}\,q(z; \theta) \leq 1 - f\left(\mathbb{E}_{\theta \sim M(\bar{S})}\,q(z; \theta)\right). \tag{31}$$

By Lemma C.1, we have for any $g : \mathbb{D} \to [0,1]$:

$$\mathbb{E}_{z \sim P}[1 - f(g(z))] \leq 1 - f(\mathbb{E}_{z \sim P}\,g(z)). \tag{32}$$

We get the sought statement in Eq. (29) by taking the expectation over $z \sim P$ of both sides in Eq. (31) and applying Eq. (32). Finally, we get the result in Eq. (30) analogously from:

$$\mathbb{E}_{\theta \sim M(\bar{S} \cup \{z\})}\,q(z; \theta) \leq 1 - f\left(\mathbb{E}_{\theta \sim M(\bar{S} \cup \{z'\})}\,q(z; \theta)\right).$$

$\qquad\square$

In Appendix E, we also derive an equivalent result with the semantics of standard generalization bounds. Moreover, in Appendix F, we show that this result immediately implies bounds on reconstruction robustness with an alternative on-average definition of baseline success rate (Guerra-Balboa et al., 2023).

We can now show the general version of our risk bound:

**Lemma 3.1.** *Suppose that $M : 2^{\mathbb{D}} \to \Theta$ satisfies $f$-DP w.r.t. either add-remove or replace-one relation. Then, for any bounded function $q : \mathbb{D} \times \Theta \to [0,1]$, any partial dataset $\bar{S} \in \mathbb{D}^{n-1}$ with $n \geq 1$, and any probability distribution $P$ over $\mathbb{D}$, we have:*

$$\mathbb{E}_{z \sim P} \, \mathbb{E}_{\theta \sim M(\bar{S} \cup \{z\})} \left[ q(z; \theta) \right] \leq 1 - f\left( \sup_{\theta \in \Theta} \mathbb{E}_{z \sim P} \left[ q(z; \theta) \right] \right). \tag{10}$$

*Proof.* For add-remove relation, we immediately get the result by applying Lemma C.2 to the r.h.s. of Eq. (29), and taking $g(\theta) = \mathbb{E}_{z \sim P} \, q(z; \theta)$:

$$\mathbb{E}_{z \sim P} \, \mathbb{E}_{\theta \sim M(\bar{S})} \, q(z; \theta) = \mathbb{E}_{\theta \sim M(\bar{S})} \, \mathbb{E}_{z \sim P} \, q(z; \theta) = \mathbb{E}_{\theta \sim M(\bar{S})} \, g(\theta).$$

Analogously, for replace-one relation, we get the result by applying Lemma C.2 to the r.h.s. of Eq. (30) with:

$$\mathbb{E}_{z \sim P} \, \mathbb{E}_{\theta \sim M(\bar{S} \cup \{z'\})} \, q(z; \theta) = \mathbb{E}_{\theta \sim M(\bar{S} \cup \{z'\})} \, \mathbb{E}_{z \sim P} \, q(z; \theta) = \mathbb{E}_{\theta \sim M(\bar{S} \cup \{z'\})} \, g(\theta).$$

$\square$

We proceed to showing that the difference between the success and the baseline is upper-bounded by TV privacy:

**Lemma C.4.** *Suppose that the algorithm $M : 2^{\mathbb{D}} \to \Theta$ satisfies $\eta$-TV privacy w.r.t. either the add-remove or replace-one relation. Then, for any bounded function $q : \mathbb{D} \times \Theta \to [0,1]$, any partial dataset $\bar{S} \in \mathbb{D}^{n-1}$ with $n \geq 1$, and any probability distribution $P$ over $\mathbb{D}$, we have:*

$$\mathbb{E}_{z \sim P} \, \mathbb{E}_{\theta \sim M(\bar{S} \cup \{z\})} [q(z; \theta)] - \sup_{\theta \in \Theta} \mathbb{E}_{z \sim P} [q(z; \theta)] \leq \eta. \tag{33}$$

*Proof.* Let us assume that $M(\cdot)$ satisfies $f$-DP for some valid trade-off curve $f$. This is always the case under the assumption of the statement, as $\eta$-TV implies $f(\alpha) = 1 - \alpha - \eta$ by Eq. (2). Denoting by $\mathsf{succ} = \mathbb{E}_{z \sim P} \, \mathbb{E}_{\theta \sim M(\bar{S} \cup \{z\})} [q(z; \theta)]$ and $\mathsf{base} = \sup_{\theta \in \Theta} \mathbb{E}_{z \sim P} [q(z; \theta)]$, recall that by Lemma 3.1:

$$\mathsf{succ} - \mathsf{base} \leq 1 - f(\mathsf{base}) - \mathsf{base}$$

To make the r.h.s. independent of the baseline, let us maximize it over all possible baseline values:

$$1 - f(\alpha) - \alpha \leq \max_{\alpha \in [0,1]} 1 - f(\alpha) - \alpha \leq \eta,$$

where the last inequality is by Proposition A.1. Therefore, $\mathsf{succ} - \mathsf{base} \leq \eta$, which is exactly the sought result.

$\square$

## C.2 Formal Versions of the Simplified Statements in the Main Body

We can now formally re-state Theorems 3.1 and 3.3:

**Theorem C.1** (Formal version of Theorem 3.1). *Suppose that the algorithm $M : 2^{\mathbb{D}} \to \Theta$ satisfies $f$-DP w.r.t. either the add-remove or replace-one relation. Then, the following hold:*

- *SPSO. For any $n \geq 1, k \geq 2$, partial dataset $\bar{S} \in \mathbb{D}^{n-1}$, data distribution $P$ over a candidate set $\mathbb{W} \subseteq \mathbb{D}$ of size $|\mathbb{W}| = k$, weight $w \in [0,1]$, and any adversary $\mathcal{A}_{M,\bar{S},P,w} : \Theta \to \mathbb{Q}_{\bar{S},P}$ for $\mathbb{Q}_{\bar{S},P} \subseteq \{p \mid p : \mathbb{D} \to \{0,1\}, \sum_{z' \in \bar{S}} p(z') = 0, \mathbb{E}_P[p] \leq w\}$, we have:*

$$\mathsf{succ}_{SPSO}(M, \bar{S}, P, w; \mathcal{A}_{M,\bar{S},P,w}) \leq 1 - f(\mathsf{base}_{SPSO}(\bar{S}, P, w)).$$

- *SRR. For any $n \geq 1$, partial dataset $\bar{S} \in \mathbb{D}^{n-1}$, data distribution $P$ over $\mathbb{D}$, loss function $\ell : \mathbb{D} \times \mathbb{D} \to \mathbb{R}$, threshold $\gamma \in \mathbb{R}$, and any adversary $\mathcal{A}_{M,\bar{S},P} : \Theta \to \mathbb{D}$, we have:*

$$\mathsf{succ}_{SRR}(M, \bar{S}, P; \mathcal{A}_{M,\bar{S},P}, \ell, \gamma) \leq 1 - f(\mathsf{base}_{SRR}(P; \ell, \gamma)).$$

- *SAI. For any $n \geq 1$, set of attributes $\mathbb{A} = \{1, \ldots, k\}$, mapping from records to attributes $a : \mathbb{D} \to \mathbb{A}$, partial dataset $\bar{S} \in \mathbb{D}^{n-1}$, data distribution $P$ over $\mathbb{D}$, and any adversary $\mathcal{A}_{M,\bar{S},P} : \Theta \to \mathbb{A}$, we have:*

$$\mathsf{succ}_{SAI}(M, \bar{S}, P; \mathcal{A}_{M,\bar{S},P}, a) \leq 1 - f(\mathsf{base}_{SAI}(P, a)).$$

*Proof.* Consider the mechanism $\mathcal{A}(M(\cdot))$, where we omit the subscripts for brevity. It satisfies $f$-DP by post-processing. Let us denote the output space of this mechanism as $\mathbb{Y}$, i.e., $\mathbb{Y} = \mathbb{Q}_{\bar{S},P}$ for SPSO, $\mathbb{Y} = \mathbb{D}$ for SRR, and $\mathbb{Y} = \mathbb{A}$ for SAI. The upper bounds follow immediately from Lemma 3.1 by taking an appropriate choice of $q : \mathbb{D} \times \mathbb{Y} \to [0, 1]$:

$$
\begin{aligned}
\text{SPSO:} \quad & q(z; p) = \mathbb{1}[p(z) = 1] \\
\text{SRR:} \quad & q(z; \hat{z}) = \mathbb{1}[\ell(z, \hat{z}) \leq \gamma] \\
\text{SAI:} \quad & q(z; \hat{a}) = \mathbb{1}[a(z) = \hat{a}]
\end{aligned}
\tag{34}
$$

$\square$

**Theorem C.2** (Formal version of Theorem 3.3). *Suppose that the algorithm $M : 2^{\mathbb{D}} \to \Theta$ satisfies $\eta$-TV privacy w.r.t. either the add-remove or replace-one relation. Then, the following hold:*

- *SPSO. For any $n \geq 1, k \geq 2$, partial dataset $\bar{S} \in \mathbb{D}^{n-1}$, data distribution $P$ over a candidate set $\mathbb{W} \subseteq \mathbb{D}$ of size $|\mathbb{W}| = k$, weight $w \in [0, 1]$, and any adversary $\mathcal{A}_{M,\bar{S},P,w} : \Theta \to \mathbb{Q}_{\bar{S},P}$ for $\mathbb{Q}_{\bar{S},P} \subseteq \{p \mid p : \mathbb{D} \to \{0, 1\}, \sum_{z' \in \bar{S}} p(z') = 0, \mathbb{E}_P[p] \leq w\}$, we have:*

$$\mathsf{succ}_{SPSO}(M, \bar{S}, P, w; \mathcal{A}_{M,\bar{S},P,w}) - \mathsf{base}_{SPSO}(\bar{S}, P, w) \leq \eta.$$

- *SRR. For any $n \geq 1$, partial dataset $\bar{S} \in \mathbb{D}^{n-1}$, data distribution $P$ over $\mathbb{D}$, loss function $\ell : \mathbb{D} \times \mathbb{D} \to \mathbb{R}$, threshold $\gamma \in \mathbb{R}$, and any adversary $\mathcal{A}_{M,\bar{S},P} : \Theta \to \mathbb{D}$, we have:*

$$\mathsf{succ}_{SRR}(M, \bar{S}, P; \mathcal{A}_{M,\bar{S},P}, \ell, \gamma) - \mathsf{base}_{SRR}(P; \ell, \gamma) \leq \eta.$$

- *SAI. For any $n \geq 1$, set of attributes $\mathbb{A} = \{1, \ldots, k\}$, mapping from records to attributes $a : \mathbb{D} \to \mathbb{A}$, partial dataset $\bar{S} \in \mathbb{D}^{n-1}$, data distribution $P$ over $\mathbb{D}$, and any adversary $\mathcal{A}_{M,\bar{S},P} : \Theta \to \mathbb{A}$, we have:*

$$\mathsf{succ}_{SAI}(M, \bar{S}, P; \mathcal{A}_{M,\bar{S},P}, a) - \mathsf{base}_{SAI}(P, a) \leq \eta.$$

*Proof.* As before, we obtain the results by applying Lemma C.4 to a mechanism $\mathcal{A}(M(\cdot))$, which satisfies $f$-DP by post-processing, and an appropriately chosen $q(z; \cdot)$ in Eq. (34). $\square$

# D  Tighter Bounds under a Bernoulli Prior

In this section, we provide a tighter bound than in Theorem 3.1 assuming a known form of the prior distribution $P$ over $\mathbb{D}$, specifically, $z = z_b$ with $b \sim \mathsf{Bern}(\pi)$ for some $\pi \in [0, 1]$. This setting can model two threats: (1) SAI of a single binary attribute with a non-uniform prior (see, e.g., Guo et al., 2023) or, equivalently, SRR with only two points in the support of the prior distribution $P$, and (2) SMIA with a non-uniform prior probability of membership (Jayaraman et al., 2021).

We make use of the notion of the Bayes error of the attacker (Chatzikokolakis et al., 2023).

**Definition D.1** (Bayes error for $f$-DP, Kaissis et al., 2024). *Suppose that $f$ is a valid trade-off function according to Definition A.3. For any $\pi \in [0, 1]$, we define the Bayes error w.r.t. the prior $\pi$ as follows:*

$$R_f(\pi) \triangleq \min_{\alpha \in [0,1]} (\pi \alpha + (1 - \pi) f(\alpha)).$$

Given a numeric representation of a trade-off curve $f$, Bayes error can be easily computed numerically by, e.g., grid search over a grid of $\alpha$ values. Additionally, we show an elegant way to obtain the Bayes error directly from the privacy profile $\delta(\varepsilon)$, which was previously unknown in the context of DP:

**Proposition D.1** (Based on Sason and Verdú, 2016, Eq. (421))**.** *Suppose that a mechanism $M(\cdot)$ satisfies $f$-DP and has an associated privacy profile $\delta(\varepsilon)$ w.r.t. any neighbourhood relation. Then, the Bayes error is bounded:*

$$R_f(\pi) \geq (1 - \pi) \cdot \left[ 1 - \delta\big(\text{logit}(\pi)\big) \right], \quad \text{where } \text{logit}(\pi) \triangleq \log\left(\frac{\pi}{1 - \pi}\right). \tag{35}$$

Next, we show a bound on success of SAI/SMIA under Bernoulli prior using the Bayes error.

**Theorem D.1.** *Fix any partial dataset $\bar{S} \in \mathbb{D}^{n-1}$ with $n \geq 1$. Suppose that $M : \mathbb{D}^n \to \Theta$ satisfies $f$-DP w.r.t. the replace-one relation. Then, for any prior probability $\pi \in [0, 1]$, a set of any two candidate records $\{z_0, z_1\} \subseteq \mathbb{D}$, and any score function $\hat{b} : \Theta \to [0, 1]$ indicating adversary's confidence of whether $z_1$ or $z_0$ was used for training, we have:*

$$\mathbb{E}_{b \sim \text{Bern}(\pi)} \mathbb{E}_{\theta \sim M(\bar{S} \cup \{z_b\})} [q(z_b; \theta)] \leq 1 - R_f(\pi), \tag{36}$$

*where we define the success indicator $q(z; \theta)$ as follows:*

$$q(z; \theta) \triangleq \begin{cases} \hat{b}(\theta), & \text{if } z = z_1 \\ 1 - \hat{b}(\theta), & \text{if } z = z_0 \end{cases}. \tag{37}$$

*Proof.* We can decompose the l.h.s. in Eq. (36) as follows:

$$\mathbb{E}_{\substack{b \sim \text{Bern}(\pi) \\ \theta \sim M(\bar{S} \cup \{z_b\})}} [q(z_b; \theta)] = (1 - \pi) \cdot \mathbb{E}_{\theta \sim M(\bar{S} \cup \{z_0\})}[q(z_0; \theta)] + \pi \cdot \mathbb{E}_{\theta \sim M(\bar{S} \cup \{z_1\})}[q(z_1; \theta)]. \tag{38}$$

Observe that in the setup of this statement, we have $q(z_1; \theta) = 1 - q(z_0; \theta)$. Denote by $\alpha = \mathbb{E}_{\theta \sim M(\bar{S} \cup \{z_0\})}[q(z_1; \theta)] = 1 - \mathbb{E}_{\theta \sim M(\bar{S} \cup \{z_0\})}[q(z_0; \theta)]$. By Lemma 2.1, we have:

$$\mathbb{E}_{\theta \sim M(\bar{S} \cup \{z_1\})}[q(z_1; \theta)] \leq 1 - f(\alpha).$$

Therefore, we can upper bound Eq. (38) as:

$$\begin{aligned} &\leq (1 - \pi) \cdot (1 - \alpha) + \pi \cdot (1 - f(\alpha)) \\ &= 1 - R_f(\pi), \end{aligned}$$

where the last equality is by Definition D.1. $\qquad \square$

**Experimental evaluation.** We compare our bounds in Theorem 3.1, which are applicable to any prior, and the specialized result in Theorem D.1 for Bernoulli priors. We additionally compare these results to the bounds on SRR based on RDP as in Section 4, and the bounds on SAI based on the Fano's inequality (Guo et al., 2023).

For this comparison, we use Gaussian mechanism with noise scales $\sigma \in \{0.5, 1.0, 2.0\}$, analyzed under replace-one relation. We use $\delta = 10^{-5}$ to compute $\varepsilon$. We use two versions of the bounds from Guo et al. (2023): (1) based on an analytical bound on mutual information (MI) between Gaussians, and a Monte-Carlo (MC) method using 10,000 samples (we repeat each run 10 times). Note that the latter method only provides an estimate of a bound, unlike all other evaluated approaches. We show the result in Fig. 5 (error bars for the MC approach are not visible). We can see that for all noise parameters, and all baselines, the bound in Theorem D.1 outperforms the other bounds.

# E  Generalization Bounds

In this section, we extend the previous results to the setting in which the dataset is assumed to be sampled i.i.d. from some data distribution $P$. The results in this section assume the replace-one neighbourhood relation.

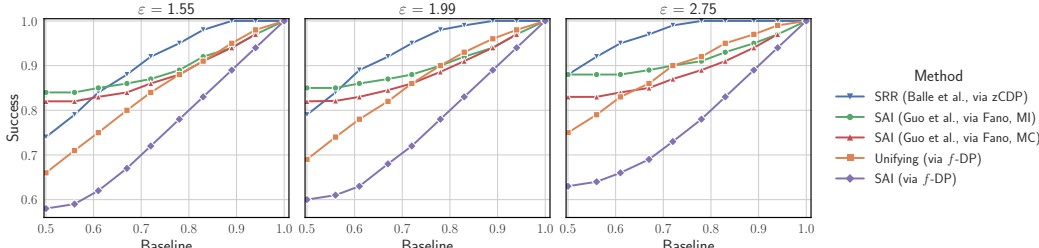

Figure 5: **We can significantly tighten the bounds in the setting of binary attribute inference (SAI, via $f$-DP), outperforming the prior bound based on Fano's inequality.**

**Theorem E.1** (On-average generalization from $f$-DP). *Suppose that $M : \mathbb{D}^n \to \Theta$ satisfies $f$-DP w.r.t. replace-one relation. Then, for any bounded loss function $q : \mathbb{D} \times \Theta \to [0, 1]$, and any probability distribution $P$ over $\mathbb{D}$, we have:*

$$\underbrace{\mathbb{E}_{\substack{S \sim P^n \\ \theta \sim M(S)}} \frac{1}{n} \sum_{i=1}^n q(z_i; \theta)}_{\text{on-average train loss}} \leq 1 - f\left(\underbrace{\mathbb{E}_{\substack{S \sim P^n, \, z \sim P \\ \theta \sim M(S)}} q(z; \theta)}_{\text{on-average test loss}}\right) \tag{39}$$

$$\underbrace{\mathbb{E}_{\substack{S \sim P^n, \, z \sim P \\ \theta \sim M(S)}} q(z; \theta)}_{\text{on-average test loss}} \leq 1 - f\left(\underbrace{\mathbb{E}_{\substack{S \sim P^n \\ \theta \sim M(S)}} \frac{1}{n} \sum_{i=1}^n q(z_i; \theta)}_{\text{on-average train loss}}\right), \tag{40}$$

*where $S = (z_1, z_2, \ldots, z_n)$.*

*Proof.* Fix any $z, z' \in \mathbb{D}$, and $\bar{S} \in \mathbb{D}^{n-1}$. By Lemma 2.1 and the post-processing property of $f$-DP, we have:

$$\mathbb{E}_{\theta \sim M(\bar{S} \cup \{z\})} q(z; \theta) \leq 1 - f\left(\mathbb{E}_{\theta \sim M(\bar{S} \cup \{z'\})} q(z; \theta)\right)$$
$$\mathbb{E}_{\theta \sim M(\bar{S} \cup \{z'\})} q(z; \theta) \leq 1 - f\left(\mathbb{E}_{\theta \sim M(\bar{S} \cup \{z\})} q(z; \theta)\right). \tag{41}$$

By using Lemma C.1 after taking the expectation over $z, z' \sim P$ and $\bar{S} \sim P^{n-1}$ of both sides in Eq. (41), we have:

$$\mathbb{E}_{\substack{S \sim P^n \\ z \sim \mathsf{Unif}[S] \\ \theta \sim M(S)}} q(z; \theta) \leq 1 - f\left(\mathbb{E}_{\substack{S \sim P^n \\ z \sim P \\ \theta \sim M(S)}} q(z; \theta)\right)$$

$$\mathbb{E}_{\substack{S \sim P^n \\ z \sim P \\ \theta \sim M(S)}} q(z; \theta) \leq 1 - f\left(\mathbb{E}_{\substack{S \sim P^n \\ z \sim \mathsf{Unif}[S] \\ \theta \sim M(S)}} q(z; \theta)\right)$$

The forms in Eqs. (39) and (40) are equivalent as $\mathbb{E}_{z \sim \mathsf{Unif}[S]} q(z; \theta) = \frac{1}{n} \sum_{i=1}^n q(z_i; \theta)$. $\qquad\square$

By applying Theorem 3.3, we recover the prior result that $\sup_{q : \mathbb{D} \times \Theta \to [0,1]} |\mathsf{err}_{\mathrm{tr}}(q) - \mathsf{err}_{\mathrm{test}}(q)| \leq \eta$ for $\eta$-TV algorithms (Kulynych et al., 2022b). Next, we show a related result which covers non-linear queries, i.e., those that do not decompose linearly over the dataset. For this, we make use of the *group privacy* property of $f$-DP:

**Proposition E.1** (Dong et al., 2022). *Suppose that $M(\cdot)$ satisfies $f$-DP w.r.t. replace-one relation. Then, the algorithm satisfies $f^{(k)}$-DP w.r.t. replace-k relation $S \simeq S'$ ($|S| = |S'|$ but differ by exactly $k$ points), where $f^{(k)} = 1 - (1 - f)^{\circ k}$, with $f^{\circ k}$ denoting repeated $k$-fold composition of the function as $f \underbrace{\circ \cdots \circ}_{k} f(x)$. The function $f^{(k)}$ is a valid trade-off curve as per Definition A.3.*

We can now show the result for non-linear queries:

**Theorem E.2.** *Suppose that $M : \mathbb{D}^n \to \Theta$ satisfies $f$-DP w.r.t. replace-one relation. Then, for any bounded function $R : \mathbb{D}^n \times \Theta \to [0, 1]$, and any probability distribution $P$ over $\mathbb{D}$, we have:*

$$\mathbb{E}_{\substack{S \sim P^n \\ \theta \sim M(S)}} R(S; \theta) \leq 1 - f^{(n)} \left( \mathbb{E}_{\substack{S, T \sim P^n \\ \theta \sim M(S)}} R(T; \theta) \right) \tag{42}$$

$$E_{\substack{S, T \sim P^n \\ \theta \sim M(S)}} R(T; \theta) \leq 1 - f^{(n)} \left( \mathbb{E}_{\substack{S \sim P^n \\ \theta \sim M(S)}} R(T; \theta) \right). \tag{43}$$

*Proof.* Fix any $S \in \mathbb{D}^n$ and $T \in \mathbb{D}^n$. By Lemma 2.1 and the group privacy property in Proposition E.1, we have:

$$\mathbb{E}_{\theta \sim M(S)} R(S; \theta) \leq 1 - f^{(n)} \left( \mathbb{E}_{\theta \sim M(T)} R(S; \theta) \right)$$
$$\mathbb{E}_{\theta \sim M(T)} R(S; \theta) \leq 1 - f^{(n)} \left( \mathbb{E}_{\theta \sim M(S)} R(S; \theta) \right). \tag{44}$$

By using Lemma C.1 after taking the expectation over $S \sim P^n$, and $T \sim P^n$ of both sides in Eq. (44), we have:

$$\mathbb{E}_{\substack{S \sim P^n \\ \theta \sim M(S)}} R(S; \theta) \leq 1 - f^{(n)} \left( \mathbb{E}_{\substack{S, T \sim P^n \\ \theta \sim M(T)}} R(S; \theta) \right)$$

$$\mathbb{E}_{\substack{S, T \sim P^n \\ \theta \sim M(T)}} R(S; \theta) \leq 1 - f^{(n)} \left( \mathbb{E}_{\substack{S \sim P^n \\ \theta \sim M(S)}} R(S; \theta) \right)$$

We get the desired result by renaming the variables $S$ and $T$ in the $\mathbb{E}_{S,T}[R(S; M(T))]$ terms. $\qquad \square$

In Appendix F, we use this result to bound the notion of narcissus resiliency (Cohen et al., 2025) using $f$-DP. Unfortunately, the privacy guarantees obtained using this approach can quickly become vacuous, as they scale with the dataset size.

# F  Bounds for Other Risk Notions

In this section, we show additional bounds based on $f$-DP for other risk notions beyond the strong-adversary notions in Section 3.2.

**Counterfactual memorization and influence.** Using the tools in Appendices C and E, we can bound notions of memorization (Feldman, 2019; Zhang et al., 2023). First, we define cross-influence:

**Definition F.1** (Cross-influence). *Fix a partial dataset $\bar{S} \in \mathbb{D}^{n-1}$, two records $z, z' \in \mathbb{D}$, and a bounded loss function $\ell : \mathbb{D} \times \mathbb{D} \to [0, 1]$. We define the cross-influence of $z'$ on $z$ as:*

$$\mathsf{xinf}(z \Leftarrow z') \triangleq \mathbb{E}_{\theta \sim M(\bar{S} \cup \{z'\})} \left[ \ell(z; \theta) \right] - \mathbb{E}_{z'' \sim P, \, \theta \sim M(\bar{S} \cup \{z''\})} \left[ \ell(z; \theta) \right]. \tag{45}$$

When $z = z'$, we obtain the special case of *counterfactual self-influence*, i.e., memorization.

**Definition F.2** (Counterfactual memorization). *Fix $\bar{S} \in \mathbb{D}^{n-1}$ to be a partial dataset, $z \in \mathbb{D}$ to be a fixed record, $P$ to be a distribution over $\mathbb{D}$, and a bounded loss function $\ell : \mathbb{D} \times \mathbb{D} \to [0, 1]$. We define the memorization of $z$ as:*

$$\mathsf{mem}(z) \triangleq \mathbb{E}_{\theta \sim M(\bar{S} \cup \{z\})} \left[ \ell(z; \theta) \right] - \mathbb{E}_{z' \sim P, \, \theta \sim M(\bar{S} \cup \{z'\})} \left[ \ell(z; \theta) \right]. \tag{46}$$

For the special case that $\ell(z; \theta)$ is a decision rule implementing a membership inference attack which aims to infer the membership of $z$, we can write the SMIA advantage in a replace-one neighbourhood model as follows:

$$\mathsf{adv}_{\mathsf{SMIA'}}(z) \triangleq \underbrace{\mathbb{E}_{\theta \sim M(\bar{S} \cup \{z\})} \left[ \phi(z; \theta) \right]}_{\text{TPR of } \phi} - \underbrace{\mathbb{E}_{z' \sim P, \, \theta \sim M(\bar{S} \cup \{z'\})} \left[ \phi(z; \theta) \right]}_{\text{FPR of } \phi}. \tag{47}$$

The next result then follows by recognizing that mem recovers the definition of $\mathsf{adv}_{\mathsf{SMIA'}}$, and that $\mathsf{adv}_{\mathsf{SMIA'}}$ is upper-bounded by the TV privacy parameter:

**Proposition F.1** (η-TV privacy bounds memorization/advantage). *Suppose that $M : \mathbb{D}^n \to \Theta$ satisfies η-TV privacy w.r.t. the replace-one neighbourhood relation. Then, for any $n \geq 1$, probability distribution $P$ over $\mathbb{D}$, partial dataset $\bar{S} \in \mathbb{D}^{n-1}$, any bounded $\ell : \mathbb{D} \times \Theta \to [0,1]$, and any target record $z \in \mathbb{D}$, it holds that:*

$$\mathsf{mem}(z) \leq \eta \text{ and } \mathsf{adv}_{SMIA'}(z) \leq \eta. \tag{48}$$

*Proof.* Recall that $M$ satisfies $f$-DP for some $f$. For any fixed $z, z' \in \mathbb{D}$, we have by Lemma 2.1:

$$\mathbb{E}_{\theta \sim M(\bar{S} \cup \{z\})} \, q(z; \theta) \leq 1 - f\left(\mathbb{E}_{\theta \sim M(\bar{S} \cup \{z'\})} \, q(z; \theta)\right),$$

Subsequently, taking the expectation over $z' \sim P$ and applying Lemma 3.1, we get:

$$\mathbb{E}_{\theta \sim M(\bar{S} \cup \{z\})} \, q(z; \theta) \leq 1 - f\left(\mathbb{E}_{z' \sim P} \, \mathbb{E}_{\theta \sim M(\bar{S} \cup \{z'\})} \, q(z; \theta)\right),$$

Finally, we obtain the bound in terms of $\eta$ by subtracting $E_{z' \sim P \; \theta \sim M(\bar{S} \cup \{z'\})} q(z; \theta)$ from both sides and recalling that $\max_{\alpha \in [0,1]} (1 - f(\alpha) - \alpha) \leq \eta$ by Proposition A.1. $\qquad\square$

Note that a similar connection between memorization and the advantage of membership inference attacks in an average-dataset threat model has been observed previously (Kulynych et al., 2022a).

**Unbiased reconstruction robustness.** Next, we present a variant of reconstruction robustness with an average baseline.

**Definition F.3** (Unbiased RR, adapted from Guerra-Balboa et al., 2023). *For a given $n \geq 1$, mechanism $M : 2^{\mathbb{D}} \to \Theta$, data distribution $P$ over $\mathbb{D}$, partial dataset $\bar{S} \in \mathbb{D}^{n-1}$, loss function $\ell : \mathbb{D} \times \mathbb{D} \to \mathbb{R}$, threshold $\gamma \in \mathbb{R}$, and reconstruction attack $\mathcal{A}_{M,\bar{S},\mathcal{D}} : \Theta \to \mathbb{D}$, we define the unbiased reconstruction robustness (URR) success rate as follows:*

$$\mathsf{succ}_{URR}(M, \bar{S}, P; \mathcal{A}, \ell, \gamma) \triangleq \Pr_{\substack{z \sim P \\ \hat{z} \leftarrow \mathcal{A}_{M,\bar{S},P}(M(\bar{S} \cup \{z\}))}} [\ell(z, \hat{z}) \leq \gamma],$$

*and the baseline success as:*

$$\mathsf{base}_{URR}(P; \ell, \gamma) \triangleq \Pr_{\substack{z, z' \sim P \\ \hat{z} \leftarrow \mathcal{A}_{M,\bar{S},P}(M(\bar{S} \cup \{z'\}))}} [\ell(z, \hat{z}) \leq \gamma].$$

**Theorem F.1.** *Suppose that the algorithm $M : \mathbb{D}^n \to \Theta$ satisfies $f$-DP w.r.t. replace-one relation. Then, for any $n \geq 1$, partial dataset $\bar{S} \in \mathbb{D}^{n-1}$, data distribution $P$ over $\mathbb{D}$, loss function $\ell : \mathbb{D} \times \mathbb{D} \to \mathbb{R}$, threshold $\gamma \in \mathbb{R}$, and any adversary $\mathcal{A}_{M,\bar{S},P} : \Theta \to \mathbb{D}$, we have:*

$$\mathsf{succ}_{URR}(M, \bar{S}, P; \mathcal{A}_{M,\bar{S},P}, \ell, \gamma) \leq 1 - f(\mathsf{base}_{URR}(P; \ell, \gamma)). \tag{49}$$

*Moreover,*

$$\mathsf{succ}_{URR}(M, \bar{S}, P; \mathcal{A}_{M,\bar{S},P}, \ell, \gamma) - \mathsf{base}_{URR}(P; \ell, \gamma) \leq \eta, \tag{50}$$

*where $\eta$ is the TV privacy guarantee of the mechanism.*

*Proof.* We obtain Eq. (49) by applying Lemma C.3 to a mechanism $\mathcal{A}_{M,\bar{S},P}(\theta)$, which satisfies $f$-DP by post-processing, and using $q(z; \cdot)$ chosen as in Eq. (34). We immediately get Eq. (50) by applying Proposition A.1 to Eq. (49) as in the proof of Lemma C.4. $\qquad\square$

**Narcissus resiliency.** We present a recent definition of privacy due to Cohen et al. (2025), adapted to our setting:

**Definition F.4** (Narcissus resiliency, adapted from Cohen et al., 2025). *For a given $n \geq 1$, mechanism $M : 2^{\mathbb{D}} \to \Theta$, data distribution $P$ over $\mathbb{D}$, an adversary $\mathcal{A}_{n,M,P} : \Theta \to \mathbb{V}$ which outputs an element from an arbitrary set $\mathbb{V}$, and a bounded function $R : \mathbb{D}^n \times \mathbb{V} \to [0,1]$, we define the narcissus resiliency (NR) success rate as follows:*

$$\mathsf{succ}_{NR}(n, M, P; \mathcal{A}, R) \triangleq \Pr_{\substack{S \sim P^n \\ p \leftarrow \mathcal{A}_{n,M,P}(M(S))}} [R(S, v)],$$

*and the baseline success as:*

$$\mathsf{base}_{NR}(n, P; \mathcal{A}, R) \triangleq \Pr_{\substack{S, T \sim P^n \\ v \leftarrow \mathcal{A}_{n,M,P}(M(S))}} [R(T, v)].$$

Our generalization result for non-linear functions in Theorem E.2 immediately implies a bound on its success:

**Theorem F.2.** *Suppose that $M : \mathbb{D}^n \to \Theta$ satisfies $f$-DP w.r.t. replace-one neighbourhood relation. Then, for any given $n \geq 1$, data distribution $P$ over $\mathbb{D}$, and an adversary $\mathcal{A}_{n,M,P} : \Theta \to \mathbb{Q}$, we have:*

$$\mathsf{succ}_{NR}(n, M, P; \mathcal{A}, R) \leq 1 - f^{(n)}(\mathsf{base}_{NR}(n, P; \mathcal{A}, R)).$$

*Proof.* After observing that $\mathcal{A}(M(S))$ satisfies $f$-DP by post-processing, the result is an immediate implication of applying Theorem E.2 to $\mathcal{A}$ as a mechanism. $\qquad\square$

**Average-dataset predicate singling out.** We can show the following bound on PSO based on $f$-DP:

**Theorem F.3.** *Suppose that $M : \mathbb{D}^n \to \Theta$ satisfies $f$-DP w.r.t. replace-one neighbourhood relation. Then, for any given $n > 1$, data distribution $P$ over $\mathbb{D}$, and an adversary $\mathcal{A}_{n,M,P} : \Theta \to \mathbb{Q}_{P,w}$ with $\mathbb{Q}_{P,w} \triangleq \{p \mid p : \mathbb{D} \to \{0,1\}, \mathbb{E}_P[p] = w\}$ for any given $w \in [0, 1/n]$, we have:*

$$\mathsf{succ}_{PSO}(n, M, P, w; \mathcal{A}) \leq n \cdot (1 - f(w)) \tag{51}$$

*Proof.* We follow the steps of the proof in [Cohen and Nissim (2020)](#). Regardless of the weight of any given predicate $p$, we can modify it to have weight less than $w$ via:

$$p^*(x) = \begin{cases} p(x) & \text{if } \mathbb{E}_P[p] \leq w \\ 0 & \text{if } \mathbb{E}_P[p] > w \end{cases}$$

Note that $p^*$ is exactly $p$ if $p$'s weight is $\leq w$, and is trivially always 0 else. So, the weight of $p^*$ is either 0 or $\mathbb{E}_P[p^*]$. Moreover, by postprocessing we know that $p^*$ is $f$-DP. We investigate the probability of a successful PSO attack on $S$, which occurs when $p^*(S) \triangleq \frac{1}{n}\sum_{i=1}^n p^*(x_i) = \frac{1}{n}$ and $\mathbb{E}_P[p^*] \leq w$. The second condition is always satisfied via the construction of $p^*$, so:

$$
\begin{aligned}
\mathsf{succ}(n, M, P; \mathcal{A}) &= \Pr_{\substack{S \sim P^n \\ p \leftarrow M(S)}} [p(S) = 1/n \text{ and } \mathbb{E}_P[p] \leq w] \\
&= \Pr_{\substack{S \sim P^n \\ p \leftarrow M(S)}} [p^*(S) = 1/n] \\
&\leq \sum_{i=1}^n \Pr_{\substack{S \sim P^n \\ p \leftarrow M(S)}} [p^*(z_i) = 1 \text{ and } \forall j \neq i, \, p^*(z_j) = 0],
\end{aligned}
$$

where $S = (z_1, z_2, \ldots, z_n)$.

Now, as $p^*$ satisfies $f$-DP, we can apply Lemma 2.1 in conjunction with Lemma C.1:

$$\leq \sum_{i=1}^n 1 - f\left(\Pr_{\substack{S \sim P^n \\ z \sim P \\ p \leftarrow M(S_{i \to z})}} [p^*(z_i) = 1 \text{ and } \forall i \neq j, \, p^*(z_j) = 0]\right),$$

where $i \to z$ denotes substitution of $z_i$ by $z$. We can reparametrize without changing the expectation:

$$= \sum_{i=1}^n 1 - f\left(\Pr_{\substack{S \sim P^n \\ z \sim P \\ p \leftarrow M(S)}} [p^*(z_i) = 1 \text{ and } \forall j \neq i, p^*(x_j) = 0]\right). \tag{52}$$

Now, we can upper bound the probability inside $f(\cdot)$ by $w$ by Lemma C.2:

$$
\begin{aligned}
&\leq \sum_{i=1}^n 1 - f(w) \\
&= n \cdot (1 - f(w)).
\end{aligned}
$$

$\qquad\square$

Note that, to tighten the bound, we slightly modified the definition of the admissible predicate in Theorem F.3 to only consider predicates of a specific weight.

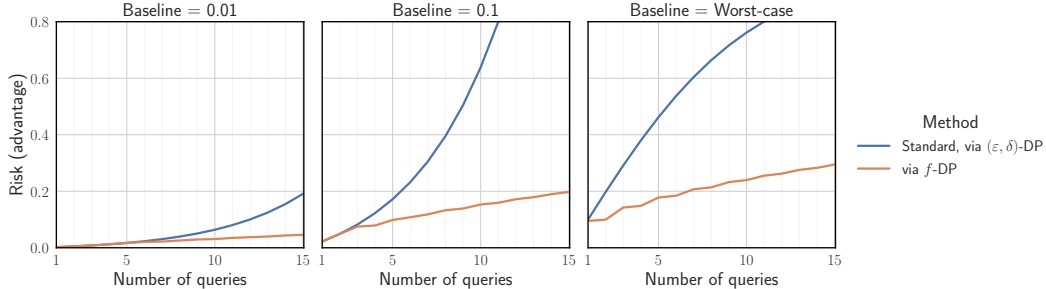

Figure 6: **Our $f$-DP based bound enables to perform more statistical queries at the same level of re-identification or other notions of risk.**

# G  Additional Experimental Details

## G.1  Additional Case Studies

**Tight budget analysis in DP query answering in terms of risk.**    One of the common use cases of DP is answering statistical queries about a dataset (Gaboardi et al., 2020). In real-world instances of such systems, e.g., in healthcare, re-identification risk is a concern (Raisaro et al., 2018).

To evaluate how our bounds apply to risk measurement in this setting, we simulate the release of multiple queries with added Laplace noise. We compare our bound to the default budget computation procedure of smartnoise software framework (Gaboardi et al., 2020), which is based on optimal composition results for $(\varepsilon, \delta)$-DP mechanisms (Kairouz et al., 2015). To apply our bounds, we use the direct method from Kulynych et al. (2024) to obtain the $f$ curve of an adaptive composition of Laplace mechanisms. In this simulation, we assume a fixed noise parameter $b = 5.0$ for each query, corresponding to $\varepsilon = 0.2$ pure DP per query. In Fig. 6, we show how our bounds on risk can enable researchers to conduct significantly more queries while satisfying any given risk requirement, in particular, on re-identification. For example, if the requirement is to ensure at most $0.2$ attack advantage under the baseline risk of $0.1$ (Fig. 6, middle pane), we can issue 15 queries according to our analysis vs. 5 with the standard one.

**Calibrating noise to image reconstruction risk in deep learning.**    In this case study, as in Section 4, we assume the modeler aims to use DP training for CIFAR-10 image classification (Krizhevsky et al., 2009) to limit such risks to a given threshold. We train multiple convolutional networks using the approach of Tramer and Boneh (2021) (we provide further technical details next). We train with standard DP-SGD (Abadi et al., 2016) and use five different levels of noise. We obtain the $f$ curve under the add-remove relation for each model using the direct method as before (Kulynych et al., 2024), and apply Theorem 3.2 to measure risk. We compare this to the RDP-based analysis from Balle et al. (2022), as described previously. Fig. 7 shows that if we aim calibrate the noise scale to a given level of maximum attack risk, our unifying bound enables to choose a lower noise scale at the same level of risk (left plot), which, in turn, results in better classification accuracy (right plot). E.g., for the risk target of $0.25$, we can increase the accuracy from 65% to 68% using our analysis, as opposed to an RDP-based one.

## G.2  Experiment Details

We use an Nvidia GeForce RTX 4070 16 GB GPU machine for the deep learning experiments. The experiments take up to four hours to finish.

**Text Sentiment Classification case study details**    We follow Yu et al. (2021) to finetune a GPT-2 (small) (Radford et al., 2019) using LoRA (Hu et al., 2021) with DP-SGD on the SST-2 sentiment classification task (Socher et al., 2013) from the GLUE benchmark (Wang et al., 2018). We summarize the parameters next:

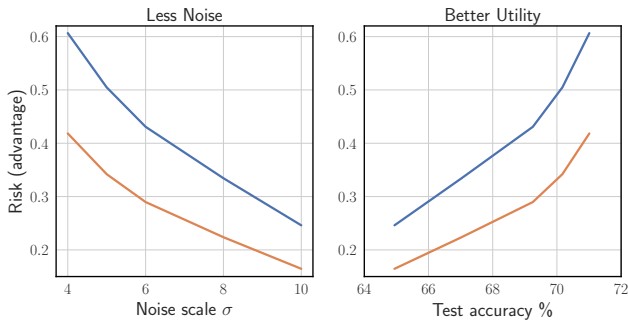

Figure 7: **Our $f$-DP based bound enables us to achieve higher classification accuracy for any level of risk.**

| Parameter | Values |
|---|---|
| Poisson subsampling probability | $\approx 0.004$ |
| Expected batch size | 256 |
| Gradient noise multiplier ($\sigma$) | $\{0.5715, 0.6072, 0.6366, 0.6945, 0.7498\}$ |
| Privacy budget ($\varepsilon$) at $\delta = 10^{-5}$ | $\{3.95, 3.2, 2.7, 1.9, 1.45\}$ |
| Training epochs | 3 |
| Gradient clipping norm ($\Delta_2$) | 1.0 |
| LoRA dimension | 4 |
| LoRA scaling factor | 32 |

**Image-classification case study details.** We use the method of Tramer and Boneh (2021) to train a convolutional neural network. We use CIFAR-10 (Krizhevsky et al., 2009) image classification dataset with a default split. We summarize the parameters next:

| Parameter | Values |
|---|---|
| Poisson subsampling probability | $\approx 0.16$ |
| Expected batch size | 8192 |
| Gradient noise multiplier ($\sigma/\Delta_2$) | $\{4, 5, 6, 8, 10\}$ |
| Training epochs | $\leq 100$ |
| Gradient clipping norm ($\Delta_2$) | 0.1 |
| Learning rate | 4 |
| Momentum (Nesterov) | 0.9 |

**Software** We use the following key open-source software:

- PyTorch (Paszke et al., 2019) for implementing neural networks.
- opacus (Yousefpour et al., 2021) for training PyTorch neural networks with DP-SGD.
- numpy (Harris et al., 2020), pandas (pandas development team, 2020), and jupyter (Kluyver et al., 2016) for numeric analyses.
- seaborn (Waskom, 2021) for visualizations.

# H   Additional Figures

Table 1: **Comparison of average-dataset vs. strong-adversary threat models.** $n$: dataset size, $P$: data distribution, $M$: mechanism, $\bar{S}$: partial dataset, *poisoning cap.:* whether an adversary has the capability to insert arbitrary records into the dataset, *: the dataset size is implicitly known from the size of the partial dataset $\bar{S}$.

| Threat model | Risk notion | Reference | Adv. knowledge | | | | Poisoning cap. | Bounds from $f$-DP |
|---|---|---|---|---|---|---|---|---|
| | | | $n$ | $P$ | $M$ | $\bar{S}$ | | |
| Strong | SPSO | Definition 3.2 | * | ✓ | ✓ | ✓ | ✓ | |
| | SRR (Balle et al., 2022) | Definition 3.3 | * | ✓ | ✓ | ✓ | ✓ | Theorem 3.2 |
| | SAI (any $k$) | Definition 3.4 | * | ✓ | ✓ | ✓ | ✓ | |
| | SAI ($k = 2$) | Definition 3.4 | * | ✓ | ✓ | ✓ | ✓ | Theorem D.1 |
| | URR (Guerra-Balboa et al., 2023) | Definition F.3 | * | ✓ | ✓ | ✓ | ✓ | Theorem F.1 |
| Avg. dataset | NR (Cohen et al., 2025) | Definition F.4 | ✓ | ✓ | ✓ | | | Theorem F.2 |
| | PSO (Cohen and Nissim, 2020) | Definition 3.1 | ✓ | ✓ | ✓ | | | Theorem F.3 |

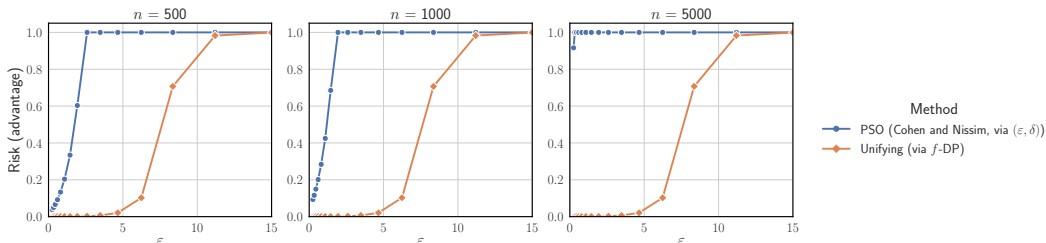

Figure 8: PSO bounds for Laplace mechanism. See Fig. 2 for details. We do not show the results with the PSO bound based on $f$-DP, as they are identical to the $(\varepsilon, \delta)$-based bound for this mechanism.

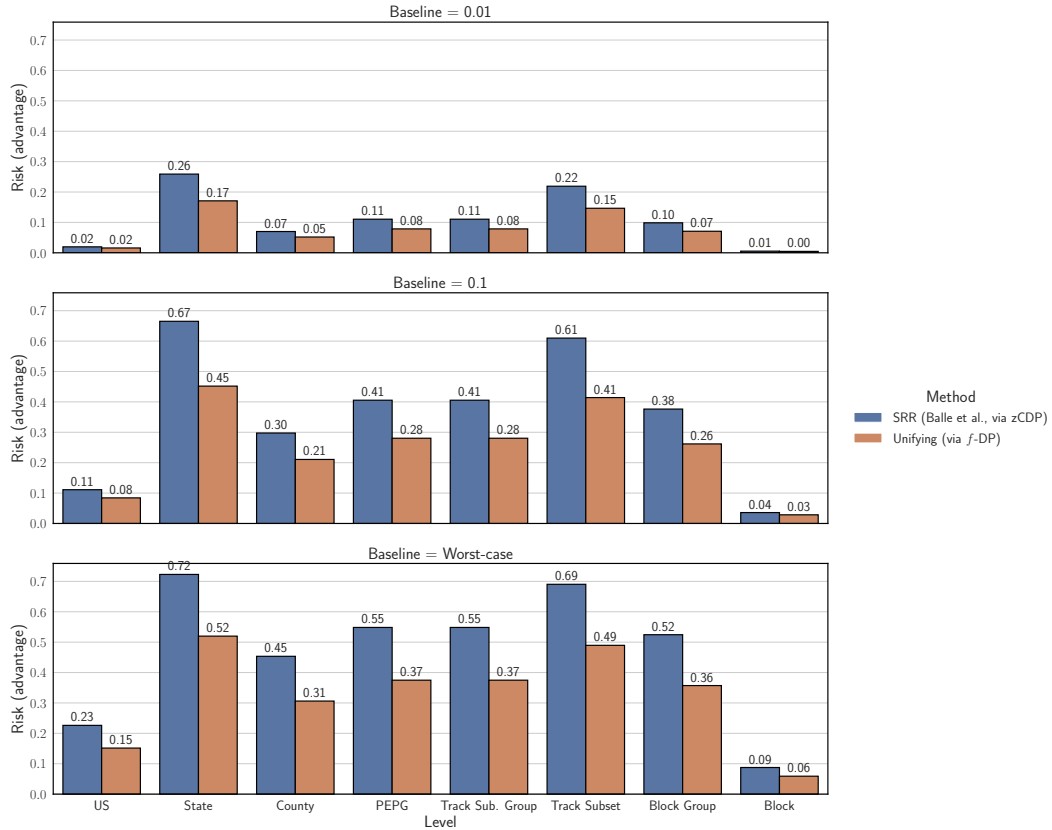

Figure 9: The results for US Census under multiple risk baselines. See Fig. 4 for details.

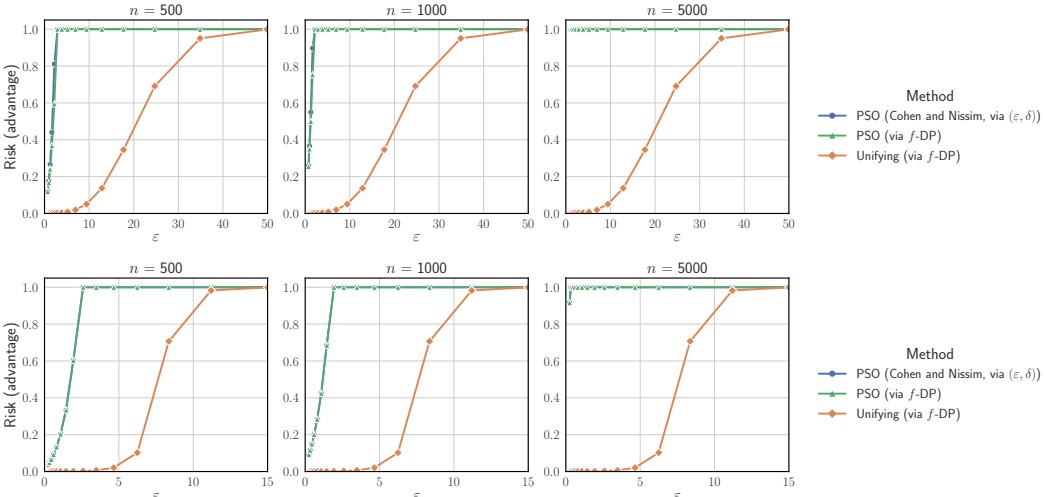

Figure 10: Result of redoing Fig. 2 (top) and Fig. 8 (bottom) with the improved $f$-DP bound for PSO derived in Theorem F.3. We observe no significant gain in using this bound.

