# OpenReview forum: "Unifying Re-Identification, Attribute Inference, and Data Reconstruction Risks in Differential Privacy"
_NeurIPS.cc/2025/Conference — NeurIPS 2025 poster_

### Official Review · Reviewer_YNGr · 2025-06-13

**Clarity:** 2
**Significance:** 3
**Originality:** 2
**Rating:** 4
**Confidence:** 5

**Summary:**

This paper introduces a framework for interpreting and calibrating DP guarantees in terms of concrete privacy risks. The main contribution is the use of the $f$-DP formalism to derive a single, unified bound on the success of re-identification, attribute inference, and data reconstruction attacks. The authors propose that for all three risks, the attack success probability is bounded by the formula $\text{succ} \le 1 - f(\text{base})$, where $\text{base}$ is the adversary's maximum success probability without the mechanism's output. The paper argues that this unifying framework provides bounds that are 1) consistent across risks, 2) tunable to specific baseline threats, and 3) empirically tighter than prior methods, which further allows for better utility.

**Questions:**

N/A

**Ethical Concerns:**

["NO or VERY MINOR ethics concerns only"]

**Final Justification:**

I believe this paper presents a solid work and will be of interest to people working on DP. I am therefore leaning towards acceptance.

**Limitations:**

No, the authors did not discuss the limitations of their work. I believe the authors should acknowledge that the bound is only applicable to risks that can be framed as a hypothesis test within the strong adversary model, but may not address other important and practical privacy risks that arise in modern machine learning.

**Quality:**

2

**Strengths And Weaknesses:**

**Strengths**
- An elegant and theoretically principled way to tightly bound different privacy risks
- Promising empirical results

**Weaknesses**

Despite its strong core idea, the paper falls short in several important issues:
- **Insufficient positioning and comparison to state-of-the-art.**
  - The paper fails to position its conceptual contribution with respect to prior work on risk-based calibration. The general idea of calibrating DP noise to a desired level of operational risk has been explored previously, in particular in [1]. While the current work uses the method from that paper as a tool, it fails to discuss the conceptual overlap and clarify what novel contributions it offers to the idea of attack-aware calibration itself.
  - The paper's central claim of providing "*tighter estimates of risk compared to prior methods*" is not backed by comparisons with the state-of-the-art for each specific risk. Most notably, for data reconstruction attacks under DP-SGD, the paper compares its bounds to Balle et al. (2022) but omits the more recent work [1], which provides nearly matching upper and lower bounds on reconstruction, suggesting their bound is already very tight. Without a direct comparison, it is impossible to assess if this paper's bound offers a meaningful improvement over the best-known specialized bound.
  - I don't think unification is a primary contribution of this paper. To me this is just a direct and unsurprising consequence of defining all risks within the same strong-adversary threat model native to DP. Even a single $(\varepsilon, \delta)$-DP guarantee can be translated into an unified upper bound ($\frac{e^{\varepsilon}+\delta}{e^{\varepsilon}+1}$). Therefore, the novelty does not lie in the unification itself, but rather in the use of $f$-DP to achieve tighter bounds.
- **Lack of motivation for the $f$-DP framework.**  The paper fails to adequately explain why the $f$-DP framework is more powerful and capable of producing tighter bounds. A critical point (in my view) is that a single $(\varepsilon, \delta)$ pair is an incomplete characterization of a mechanism's privacy guarantees. By compressing a full privacy profile into one point, prior analyses lose information, leading to looser bounds. This is in contrast to the $f$-DP framework based on the trade-off curve, which provides a complete characterization. This motivation is critical and should be a more central part of the paper's narrative.
- **Overstatement.**  The authors overstate the generality of their results by claiming that "*By bounding the success of the strong adversary, we automatically bound the success of attacks under any other threat model that is weaker or more realistic*". This claim is only valid for threat models that are strict subsets of the strong adversary model. However, many practical privacy risks, such as training data extraction (or verbatim memorization) from language models, operate under entirely different adversarial assumptions; DP and $f$-DP might not easily translate to an upper bound for these attacks (see [2]). These threat models are not necessarily "weaker", but rather orthogonal. The paper should be more precise and qualify its claims, acknowledging that its framework provides tight bounds for risks that can be framed as a hypothesis test within the strong adversary model, but may not address other important, practical privacy risks that fall outside this specific formulation.
- **Clarity of presentation and notation.** The paper's notation is dense and its definitions of risk are difficult to parse without significant effort. The concept of "baseline success" is not built up with sufficient intuition in the main body. For the work to be broadly accessible and impactful, a clearer, more streamlined presentation is needed to explain what the baseline represents and how a practitioner should reason about it.


**References**

[1] Kulynych, Bogdan, et al. "Attack-aware noise calibration for differential privacy." Advances in Neural Information Processing Systems 37 (2024): 134868-134901.

[2] Hayes, Jamie, Borja Balle, and Saeed Mahloujifar. "Bounding training data reconstruction in dp-sgd." Advances in neural information processing systems 36 (2023): 78696-78722.

[3] Hu, Yuzheng, et al. "Empirical Privacy Variance." arXiv preprint arXiv:2503.12314 (2025).

---

> ### Author Rebuttal · Authors · 2025-07-30
>
> Thank you for the review and for the detailed feedback. We address your comments and questions grouped by topic.
>
> ______
>
>
> ## The relation to the idea of attack-aware calibration
>
> > The paper fails to clarify what novel contributions it offers to the idea of attack-aware calibration itself
>
> To clarify this, we will add the following in Section 5 (Related Work):
>
>
> _Kulynych et al., 2024 introduced _attack-aware calibration_, the idea of calibrating DP algorithms to a given level of operational attack risk in contrast to the standard practice of calibrating to a given pair of $(\varepsilon, \delta)$ parameters. They evaluated this approach for the use case of calibrating noise to a given level of membership inference risk. One of the applications of our result is that it extends the toolbox of attack-aware calibration to further attack risks, thus enabling practitioners to calibrate DP algorithms to notions of risk beyond membership inference, which often appear in data protection guidelines (see Sec. 1)._
>
> _______
>
> ## The relation to Hayes et al. (2024)
>
> > the paper compares its bounds to Balle et al. (2022) but omits the more recent work [1], which provides nearly matching upper and lower bounds on reconstruction.
>
> We assume there was an incorrect reference here and the reviewer meant Hayes et al. (2024), as we were unable to find lower bounds in [1].
>
> There are several core differences of our unified bound results to the method introduced by Hayes et al: (1) they provide a Monte-Carlo sampling-based method rather than a provable bound, (2) their method is specific to DP-SGD, and (3) it only covers reconstruction attacks. In contrast, our work focuses on the fundamental connection between different standard types of risks and the $f$-DP framework, as opposed to providing an approximate method to evaluate a specific type of privacy risk in DP-SGD. Our approach enables us to obtain _provable bounds on risks for general DP mechanisms_. Even though the theoretical underpinnings of our approaches are different, the outputs of the algorithm from Hayes et al. will asymptotically (in the limit of infinite Monte-Carlo samples) converge to our $f$-DP bound in the setting of reconstruction risk in DP-SGD.
>
> We will make this clear by adding the above in our discussion of that work in Sec. 5 (Related Work).
>
>
> ________
>
> ## Limitations of the strong-adversary threat model
>
> > The authors overstate the generality of their results by claiming that _"By bounding the success of the strong adversary, we automatically bound the success of attacks under any other threat model that is weaker or more realistic"_
>
> This is a legitimate concern! There certainly are risks to the privacy of individuals whose information is present in the training data that could still exist under DP when it is not an appropriate tool to ensure privacy protection. As the reviewer pointed out, this could be the case when training on public text data from the internet (see, e.g., [Brown et al, 2022](https://arxiv.org/pdf/2202.05520)), where it is not feasible to meaningfully delineate the privacy unit.
>
> Importantly, however, the cited sentence does _not_ overstate the claims of our paper, but rather _inaccurately describes the implications of the standard strong-adversary threat model._ What we wanted to say there is that under the basic assumption – that the privacy unit and, therefore, the neighbourhood relation, are meaningful – the success of the strong-adversary MIA must bound in some way the relative increase in success of _any possible attack_ against the considered privacy unit due to the release of the mechanism outputs. This is a consequence of the fundamental decision-theoretic result known as Blackwell’s informativeness theorem (see, e.g., Theorem 2.10 in [Dong et al., 2022](https://arxiv.org/pdf/1905.02383) for its application to DP).
>
> We will make the following changes to make the description more precise. First, we will add the clarifications (in bold) in L84:
>
> _We say that two datasets $S, S’$ are neighbouring if they belong to a neighborhood relation denoted as $S \simeq S’$. **To capture meaningful concerns in privacy of the training data, this relation must correspond to adding, removing, or replacing all of the data that belongs to a single _privacy unit_, e.g., individual or secret. In the rest of the paper, we follow the standard practice and assume that the privacy unit corresponds to a single record.**_
>
> and in L136:
>
> _By bounding the success of the strong adversary, we **also bound the success of attacks against the considered privacy unit that rely on the mechanism output $M(S)$ under other threat models that are weaker or more realistic.**_
>
> > I believe the authors should acknowledge that the bound is only applicable to risks that can be framed as a hypothesis test
>
> We agree it’s a good call to acknowledge the fundamental limitations of the protection that DP can provide. As we explained before, however, reduction to a hypothesis test is a universal property of any notion of relative success of an attack against the privacy unit due to the mechanism output, as an implication of Blackwell’s informativeness theorem. We propose the following addition to Section 6, which we believe is more accurate:
>
> _**Fundamental limits of DP.** Our bounds on the relative increase in attack success are meaningful only insofar that DP is an appropriate tool and is implemented correctly. For instance, in some applications such as training of language models on open internet data, it might not be practically feasible to define a neighborhood relation in a way that meaningfully captures privacy of individuals, as the sensitive data of any one individual can appear in different forms across multiple documents (see, e.g., [Brown et al, 2022](https://arxiv.org/pdf/2202.05520))._
>
> ________
>
> ## Clarity
>
> > The concept of “baseline risk” is not built up with sufficient intuition
>
> This is a good point. We will add additional intuition in Sec. 3.4 (Tunability). As an example, consider releasing outputs of a mechanism trained on tabular medical data which contains a column corresponding to a patient’s HIV status. In this case, we might want to evaluate the risk of an adversary inferring the HIV status under standard threat models considered when releasing medical data ([El Emam, 2010](https://ieeexplore.ieee.org/abstract/document/5470957)) such as _“marketer”_ or _“journalist”_. For instance, the baseline risk of a ”marketer” adversary – who does not target any specific individual – could be modeled as guessing based on the prevalence of HIV in the population. The baseline risk of a ”journalist” adversary – who is assumed to be able to obtain side information on the target from public sources – could be modeled as guessing based on a target patient’s demographics.
>
> We note that the baseline dependence is not unique to our work, and is in fact fundamental if one cares about obtaining tight bounds on risk. Prior work both on singling out and reconstruction robustness uses notions that are equivalent to baseline risks.
>
> > The paper's notation is dense
>
> For all the notions of risk we follow the notation from prior work, and, arguably, slightly simplify the notation on singling out. Moreover, we simplify the presentation of our main results, deferring the exact statements to App. C.2 to focus on the high-level ideas in the main body. We are happy to consider any suggestions on simplifying the notation.
>
> ________
>
> ## On whether our main contribution is unification or the usage of $f$-DP
>
> We agree that the use of $f$-DP is a significant component of our contributions. We would like to highlight, however, that in our work, the unification and the usage of $f$-DP are quite interconnected.
>
> >  I don't think unification is a primary contribution of this paper.
>
> We do not unify the different types of risk _just for the unification’s sake_ (even though it makes the bounds convenient to use in practice). The primary benefit is that by unifying them _under the framework of $f$-DP,_ we provide _substantially_ more precise analyses of risk…
>
>
> > Lack of motivation for the f-DP framework.
>
> …and the reason why our bounds are tighter is exactly because we use f-DP and modern precise accounting techniques (see L219). We explicitly mention that this improvement is due to f-DP in L61 of the Introduction. As the prior work has already shown that f-DP enables tight bounds on operational notions of risk [1], we did not dwell on this point further.
>
> We agree that we should further highlight the role of f-DP. In particular, we are thinking to add the following statement to the conclusions:
>
>
> _**Decision-theoretic analyses of DP mechanisms.** Our empirical and theoretical results demonstrate that the decision-theoretic view of DP via the $f$-DP framework is a useful tool for analyzing privacy-preserving algorithms in terms of operational risks. As we have shown, it enables both substantially tighter characterizations of privacy risk compared to other standard ways to quantify leakage such as Rényi DP or CDP, and easy-to-use unified analyses of various types of privacy risk._
>
> _________
>
> Let us know if we can clarify anything else.

---

> > ### Comment · Reviewer_YNGr · 2025-08-01
> >
> > This is a high-quality rebuttal that not only addresses my concerns and comments, but also clearly marks the changes the authors plan to incorporate in their revision based on my feedback. I am therefore increasing my score.

---

### Official Review · Reviewer_GSiV · 2025-06-29

**Clarity:** 3
**Significance:** 3
**Originality:** 3
**Rating:** 4
**Confidence:** 3

**Summary:**

A problem when evaluating the privacy risks is that there is not a unified privacy risk calibrating framework for different type of threats, i.e., re-identification, attribute inference, etc. In this paper, the authors borrow f-dp to build an unified privacy-risk evaluation framework for these risks.

**Questions:**

To my understanding, you design a new risks bounding mechanism which takes the noises, algorithms, baseline success rate as inputs and outputs the theoretical attacking advantage. Is my understanding correct?

**Ethical Concerns:**

["NO or VERY MINOR ethics concerns only"]

**Final Justification:**

The response addresses most of my concerns. I think unifying the different types of risks is meaningful for the practical usage of DP. For instance, it's possible to more tightly estimate the practical privacy risk behind training data and thus value the data privacy.

**Quality:**

3

**Strengths And Weaknesses:**

Strength:

1. The problem itself is interesting.


Weakness:

1. The presentation is not clear enough to follow. For instance, in figure 1, what does $\frac{\text{re-identification}}{\text{attribute inference}}$ mean? Does it mean re-identification \textbf{or} attribute inference? And what does the f(x) mean in this figure?

I also recommend the author present a notation table to unify the symbols used in the paper.

2. Please convince me the practical reason and need of unifying the different type of risks. To my understanding, membership inference attacks, which are the most easy attacks, is proper to evaluate the privacy risks. For instance, other high-level attacks, e.g., inversion attacks, are harder to apply than membership inference attacks if the adversary has the same level of ability. In this case, if we observe an low membership inference, other type of risks should be smaller. We only need to bound the membership inference risks, why we need to bound other type of risks?

---

> ### Author Rebuttal · Authors · 2025-07-30
>
> Thank you for the review. In this rebuttal, we explain:
> - That the problem of quantifying privacy risk in terms of notions of risk that are not membership inference is important in many practical applications.
> - That we have extensively shown with experiments in Sec. 4 and App. G that the unification of different notions of risks _under $f$-DP_ enables one to obtain better utility at the same level of risk in DP, sometimes increasing accuracy by double digits.
>
> We detail on these and other points raised in the review next.
> ________
>
> ### MIAs vs. other risks
>
> > To my understanding, membership inference attacks, which are the most easy attacks, is proper to evaluate the privacy risks.
>
> In data-protection practice, membership inference is far from the main threat considered. For instance, the guidelines from ISO and European Medicines Agency mention re-identification/singling out and inference risks (see L33 in the Introduction), and, as a salient example, in the medical domain, stakeholders are most likely to be concerned about data reconstruction rather than membership inference ([Ziller et al., 2025](https://www.nature.com/articles/s42256-024-00858-y)). Moreover, the notion of reconstruction robustness is also standard in the literature on ML privacy (there are 200+ citations of the work by [Balle et al., 2022](https://arxiv.org/abs/2302.07225) who introduced the definition we use).
>
> >  if we observe an low membership inference, other type of risks should be smaller
>
> Your intuition is correct, but this fact has not been formalized in a precise way previously. Indeed, that all of the standard risk notions (singling out, attribute inference, data reconstruction) can be bounded with the error rate of a binary decision problem which is equivalent to a success rate of the strong-adversary membership inference attack, _(1) is non-trivial, (2) has not been known prior to our work, and (3) is exactly our main result._
>
> > Please convince me the practical reason and need of unifying the different type of risks.
>
> We do not unify the different types of risk just for the unification’s sake (even though it makes the bounds convenient to use in practice). The primary benefit is that by unifying them under the framework of f-DP, we can provide _substantially_ more precise analyses of risk. Where prior techniques might have indicated that a given algorithm provides no privacy protection whatsoever, our analysis could indicate very low risk (e.g., compare red and orange in Fig. 1 for an example of an analysis of singling out risk). Therefore, if one were to use a prior technique to decide whether data or model sharing is safe, in that case they would likely decide against it and possibly had applied more noise, further hurting utility. Because our analyses provide substantially more precise risk analyses, our method enables data and model sharing with high utility in tightly regulated settings, e.g., in healthcare or financial domains, where data and model releases must not exceed a certain level of privacy risk.
>
>
> ________
>
> ### Presentation
>
> > Presentation issues in Figure 1.
>
> $f(x)$ is exactly the $f$ in f-Differential Privacy (see the second sentence of the caption and Sec. 2 for the definition). We realize that the stack in the figure could be interpreted as a ratio, which we will fix by changing the line style. The stack signifies the fact that in both the success term (top) and the baseline term (bottom) we can pick any of the three standard types of privacy risk. If you have any suggestions on improving this illustration, we would be very happy to consider them.
>
> > I also recommend the author present a notation table.
>
> Absolutely, we will add one to the appendix.
>
>
> __________
>
> ### Questions
>
> > To my understanding, you design a new risks bounding mechanism which takes the noises, algorithms, baseline success rate as inputs and outputs the theoretical attacking advantage. Is my understanding correct?
>
> Yes. To be a bit more precise, given those, our method outputs maximum attainable attack advantage due to mechanism release within the strong-adversary threat model (see Theorem 3.3). Experimentally, we also show that this method is substantially more precise than existing standard methods to conduct similar analyses, and that it enables to attain significantly higher utility when calibrating the algorithm’s parameters (noise) to a given level of risk.
>
> ___________
> Let us know if we can clarify anything else.

---

> > ### Comment · Reviewer_GSiV · 2025-08-05
> >
> > Thank you for your detailed response; it addresses nearly all of my concerns, so I will raise my score accordingly.
> >
> > I do have one broader question about privacy that has puzzled me for some time.
> >
> > In research, we often measure privacy by estimating the probability that a particular sample—or certain attributes—can be re-identified. If the re-identification risk is low, we consider the sample relatively private. In everyday practice, however, people equate privacy with specific forms of personally identifiable information (PII), such as ID numbers or facial images. Would it be feasible—and meaningful—to quantify these concrete PII elements and integrate them into the current privacy-risk framework? I would value your thoughts about this question.

---

> ### Comment · Reviewer_YNGr · 2025-08-05
>
> There are two lines of research in machine learning privacy. One (which the current paper falls under) focuses on the theoretical side, particularly on DP and its variants, with the goal of deriving provable bounds on worst-case privacy risks. This line typically assumes strong threat models (e.g., the adversary knows all but one sample). The other line (which seems to be what Reviewer GSiV was thinking about) is more empirical. It aims to measure memorization or extraction of sensitive information such as PII, often in realistic settings (see, e.g., [1, 2]). Here, the goal is to quantify how much private information can be inferred or extracted in practice.
>
> To the best of my understanding, these two lines of work are not easily reconcilable (see, e.g., [3]). The primary challenge lies in their fundamentally different threat models. The first requires a well-defined privacy unit (typically a single sample or attribute), which is straightforward for tabular data but much less so for text or images. In contrast, the second often assumes partial knowledge of the target (e.g., the area code of a phone number) when evaluating (verbatim) memorization, a scenario that falls outside the assumptions of the first model.
>
> I believe it would be helpful for the authors to clarify that their framework aims to unify various privacy attacks within the first strong threat model, but does not directly address privacy risks under the second, more practical, and more open-ended setting.
>
> [1] Carlini, Nicholas, et al. "The secret sharer: Evaluating and testing unintended memorization in neural networks." 28th USENIX security symposium (USENIX security 19). 2019.
>
> [2] Carlini, Nicholas, et al. "Quantifying memorization across neural language models." The Eleventh International Conference on Learning Representations. 2022.
>
> [3] Hu, Yuzheng, et al. "Empirical Privacy Variance." arXiv preprint arXiv:2503.12314 (2025).

---

> > ### Comment · Reviewer_GSiV · 2025-08-06
> >
> > Thanks for your opinion. I have no further questions.

---

### Official Review · Reviewer_7rsk · 2025-06-30

**Clarity:** 2
**Significance:** 3
**Originality:** 2
**Rating:** 4
**Confidence:** 3

**Summary:**

This paper derives an upper bound on the success probability of any strong adversary against an $f$-DP mechanism. The bound is general—applicable to reconstruction, attribution, and other attack types—and is expressed relative to a chosen baseline. This formulation offers flexibility: for high-risk models one can adopt a universal worst-case baseline, while for lower-risk settings a more specific baseline yields a tighter bound. Empirically, the authors demonstrate that their bound strictly improves on existing Rényi-DP-based bounds for reconstruction attacks.

**Questions:**

Can the authors please elaborate on whether the proposed baseline‐dependent bound can be used to empirically audit differential privacy, and if so, how this would be carried out?

Additionally, in many settings, the f function relies on numerical procedure to compute and does not have close form solution, does that affect the interpretation of the results?

**Ethical Concerns:**

["NO or VERY MINOR ethics concerns only"]

**Final Justification:**

I believe this paper addresses a practical question often raised in DP applications and provide a way to interpret DP parameter in terms of different attack success rates. Their theoretical bounds are tighter than existing ones.

My main concern— the efficiency and accuracy of the f-DP accountant on more complex mechanisms (and their compositions)— is addressed in the rebuttal. Therefore, I maintain my score and lean toward acceptance.

**Limitations:**

Yes

**Quality:**

2

**Strengths And Weaknesses:**

**Strengths**

- The proposed upper bound sheds light on how to DP parameters to defend against specific privacy attacks, a key obstacle to the practical adoption of differential privacy.
- The authors derive an upper bound on any adversary’s success rate and demonstrate that this bound is tighter than existing reconstruction‐attack bounds.

**Weaknesses**

- The f-DP bound may not extend straightforwardly to mechanisms beyond simple Laplace or Gaussian noise; for example, its tightness under composition (as in DP-SGD) is unclear.
- Although the bound gains flexibility by depending on a baseline attack, how to select an appropriate baseline remains an open question.

---

> ### Author Rebuttal · Authors · 2025-07-30
>
> Thank you for the review. The review highlighted two weaknesses:
> 1.  _Tightness and applicability of $f$-DP analyses is unclear._ In the response, we provide evidence that the tools for analyzing mechanisms in terms of $f$-DP are reliable, precise, and are applicable to many if not most practical DP algorithms in ML.
> 2. _It is unclear how to select the baseline._ We will add examples for how one could select baselines when evaluating risk using our framework in the manuscript. We note that baseline dependence is not unique to our work, and prior work both on singling out and reconstruction robustness uses notions that are equivalent to baseline risks.
>
> We provide details next.
> _____
> ## Tightness and applicability of $f$-DP accounting
>
> > The f-DP bound may not extend straightforwardly to mechanisms beyond simple Laplace or Gaussian noise
>
> This concern is not well-founded, and we explain why next. The trade-off curve is theoretically equivalent to the privacy profile $\varepsilon(\delta)$, i.e., the set of all attainable $(\varepsilon, \delta)$ parameters, and these two representations can be losslessly converted between each other in practice via convex conjugation (see [Dong et al., 2022](https://arxiv.org/abs/1905.02383)). Recent privacy accounting tools provide tight privacy profiles for a large variety of practical mechanisms under adaptive composition, including DP-SGD.
>
> In particular, we use the Connect-the-Dots accountant ([Doroshenko et al. 2022](https://arxiv.org/abs/2207.04380)) which can compute precise privacy profiles for arbitrary adaptive compositions of Gaussian, Discrete Gaussian, Subsampled Gaussian, Mixture-of-Gaussians, Laplace, Discrete Laplace, and Randomized Response mechanisms. This covers a large class of DP algorithms.
>
> While some notable DP algorithms cannot yet be analyzed with state-of-the-art accountants, e.g., PATE ([Papernot et al., 2017](https://arxiv.org/abs/1610.05755)) or AIM ([McKenna et al., 2022](https://arxiv.org/abs/2201.12677)), even in these cases, our bound is applicable through a known conversion from Rényi DP or zCDP to $f$-DP (e.g., [Zhu et al., 2021](https://arxiv.org/abs/2106.08567)).
>
> > for example, its tightness under composition (as in DP-SGD) is unclear.
>
> It is well-established in prior work, e.g., [Nasr et al., 2023](https://arxiv.org/abs/2302.07956) that modern accountants provide analyses that are essentially tight in the strong threat models. We will clarify by extending and rearranging the discussion with the text in bold in Sec 3.4:
>
> _For more complex, e.g., composed, algorithms such as DP-SGD (Abadi et al., 2016), there are two ways to obtain the corresponding $f$ curve. First, we can estimate it using Eq. (2) from the privacy profile of the algorithm (Balle et al., 2018), i.e., the set of all attainable ($\varepsilon, \delta)$-DP pairs. This method has been used for privacy auditing previously (Nasr et al., 2023). Second, it is possible to analyze DP-SGD or other algorithms that are compositions of (subsampled) Gaussian and Laplace mechanisms using a direct method (Kulynych et al., 2024). **Both approaches provide tight analyses when using state-of-the-art accounting tools as a backbone, such as the Connect-the-Dots accountant (Doroshenko et al. 2022). In particular, prior work (Nasr et al., 2023) has shown that the upper bounds on privacy leakage obtained with the modern accounting techniques (see, e.g., Gopi et al., 2021; Alghamdi et al., 2022; Doroshenko et al., 2022) can be nearly reached by empirical attacks in the strong-adversary MIA threat model.**_
>
> > the $f$ function relies on numerical procedure to compute and does not have close form solution, does that affect the interpretation of the results?
>
> The methods described above guarantee that the numerically estimated $\tilde f$ under composition is correct, that is, it is guaranteed that $\tilde f$ is a tight lower bound on the the trade-off curve for the mechanism (Doroshenko et al., 2022, Kulynych et al., 2024). Thus, the fact that the $f$ curves are numerically computed does not affect the correctness of the results or the safety of the mechanisms, up to numerical precision.
>
> _______
> ## Choosing baseline risk
>
> > Although the bound gains flexibility by depending on a baseline attack, how to select an appropriate baseline remains an open question.
>
> This is a fair point, however we believe that the tunability of our bounds based on the baseline probability is a strength, not a weakness. Our results enable practitioners to obtain a more precise risk assessment when such a baseline estimate is available, or use the baseline-independent bounds otherwise. We will add additional intuition on how to estimate the baseline in Sec. 3.4 (Tunability).
>
> As an example, consider releasing outputs of a mechanism trained on tabular medical data which contains a column corresponding to a patient’s HIV status. In this case, we might want to evaluate the risk of an adversary inferring the HIV status under standard threat models considered when releasing medical data ([El Emam, 2010](https://ieeexplore.ieee.org/abstract/document/5470957)) such as _“marketer”_ or _“journalist”_. For instance, the baseline risk of a ”marketer” adversary – who does not target any specific individual – could be modeled as guessing based on the prevalence of HIV in the population. The baseline risk of a ”journalist” adversary – who is assumed to be able to obtain side information on the target from public sources – could be modeled as guessing based on a target patient’s demographics.
>
> This approach could also be extended to having a distribution over the possible baselines, as studied in prior work in the context of Bayesian interpretation of $f$-DP ([Kaissis et al., 2024](https://arxiv.org/abs/2406.08918)).
>
> > Can the authors please elaborate on whether the proposed baseline‐dependent bound can be used to empirically audit differential privacy, and if so, how this would be carried out?
>
> Absolutely. One would compare an empirical estimate of attack success (e.g., with a Clopper-Pearson confidence interval if the attack success is binary) against the baseline risk. This is a common approach in auditing (see, e.g., [Nasr et al., 2021](https://arxiv.org/abs/2101.04535)). As in auditing we control the canary addition to the training data set, we could sample the canaries according to a specially constructed prior distribution that would enable easy analytical computation of baseline risk, e.g., uniformly from a discrete set $P \sim \mathsf{Unif}\\{ c_1, c_2, \ldots, c_k \\}$, in which case the baseline risk of, e.g., an exact reconstruction attack, is $\mathsf{base} = 1/k$. In this case, if we observe that the Clopper-Pearson  $(1 - 2\alpha)$-lower confidence bound of the success of exact reconstruction is greater than $1 - f(\mathsf{base})$, then we can reject the hypothesis that DP is implemented correctly at significance $\alpha$.
> ______
>
> We are happy to clarify any further points.

---

> > ### Comment · Reviewer_7rsk · 2025-08-06
> >
> > Thank you for your response. It addresses my concerns.

---

### Official Review · Reviewer_gnWx · 2025-07-02

**Clarity:** 3
**Significance:** 3
**Originality:** 3
**Rating:** 5
**Confidence:** 4

**Summary:**

This paper proposed an unified view for the success of reconstruction/re-identification/attribute inference attacks through f-DP. The key technical insight is that the properties of f-DP, most importantly post-processing and convexity of the trade-off function, can be used to form a single upper bound on the success of all the aforementioned attacks based on the attacker's baseline success probability. Authors show numerically that the proposed bound provides a more meaningful upper bound than the earlier works.

**Questions:**

1. Authors could further discuss the connection on the attacks to the DP threat model. In the various security definitions, the unknown sample $z$ is averaged over, while in DP threat model we consider the threat as the supremum of the privacy loss over the $z$. This might make it easier to see the exact connection between the protection against MIA that (f-)DP provides and the threats considered in the attacks.
2. Authors could add a more detailed description of the singling-out attack. It is somewhat difficult to parse from Definition 3.1., so some general description would make it easier to follow.

**Ethical Concerns:**

["NO or VERY MINOR ethics concerns only"]

**Final Justification:**

Authors response addresses all my concerns. I'm especially happy with the proposed clarity update regarding the threat model discussion, which I expect authors to include in the revised version of the paper. I retain my original positive evaluation that the paper presents an interesting and novel perspective on unifying various notions or privacy threats.

**Limitations:**

yes

**Quality:**

3

**Strengths And Weaknesses:**

## Strengths
- Deriving tight translations of DP (or f-DP guarantees) to attacks other than membership inference is an important task. With a better characterization of the privacy threats, analyst can better calibrate their noise levels for the private release. As an example of this, authors demonstrate how you can obtain the same operational privacy risk but with significantly reduced noise compared to the prior work. (**Significance + Originality**)
- The proposed theorems give simple expressions for upper bound for the attack success, only dependent on the baseline vulnerabilities. (**Clarity + Quality**)

## Weaknesses
- Some empirical experiments on attacks would have strengthened the paper, allowing the reader to better understand the gap between the upper bounds and actual real-world risks. (**Significance**)

---

> ### Author Rebuttal · Authors · 2025-07-30
>
> Thank you for the review and for the positive assessment of our work. We address your questions and comments next.
>
> _____
>
> > Some empirical experiments on attacks would have strengthened the paper, allowing the reader to better understand the gap between the upper bounds and actual real-world risks.
>
> We could not run new evaluations for this rebuttal period, but prior work in DP auditing has extensively studied the tightness of the theoretical bounds w.r.t. empirical attacks, and showed that modern accountants such as those that we rely on provide analyses that are essentially tight in the strong threat models. However, within weaker threat models it is unlikely that these bounds are reached. We are thinking of adding the following in Section 6 (Concluding remarks) to clarify this:
>
> _**Practical tightness of the bounds.** Prior work (see, e.g., Nasr et al., 2023) has shown that the $f$-DP curves obtained using state-of-the-art accounting techniques are nearly tight in the worst-case threat models, i.e., we can construct worst-case attacks whose success rates match the theoretically predicted success rates. In more realistic threat models, attacks are unlikely to reach these success rates. Obtaining analyses of risks under relaxed threat models (see, e.g., [Kaissis et al. 2023](https://openreview.net/pdf?id=BRSgVw85Mc), [Swanberg et al., 2025](https://arxiv.org/abs/2507.08158)) is an important direction for future work._
>
> _____
>
> > Authors could further discuss the connection on the attacks to the DP threat model. In the various security definitions, the unknown sample is averaged over, while in DP threat model we consider the threat as the supremum of the privacy loss over.
>
> This is a good point. The DP threat model is slightly different from the strong-adversary threat model that we consider. In our model, the adversary has a prior distribution over the target record $z$, whereas in the DP threat model the adversary has the target record $z$. Moreover, it was indeed implicit in the discussion of our threat model that the adversary’s goal is to construct an attack with a high success rate on average over the prior. To clarify this, we will add the following (bold) in Sec. 3.1 (Threat model):
>
> _We focus on the risk within the strong threat model in which the adversary has access to the workings of the privacy-preserving algorithm, has access to the partial dataset except for one target record, and has side information about the remaining element in the form of a prior distribution. **Note that the difference with the standard threat model of SMIA (Sec. 2) is that we do not assume that the adversary knows the target record. To quantify the adversary’s success rate, we consider expectations over the prior distribution as well as over the randomness of the algorithm, as we detail next.**_
>
> Taking a supremum over the prior is also a possibility, and should yield a meaningful notion of risk that is still prior-dependent (however, only through the support of the prior distribution), but more pessimistic. This could be related to the notion of maximal leakage ([Issa et al., 2018](https://arxiv.org/abs/1807.07878)), or, equivalently, Sibson’s mutual information, between $X$ and $Y$, where $X \sim P$ and $Y \sim M(\bar S \cup \\{X\\})$. Indeed, maximal leakage has an attack interpretation, also depends on $X$ only through its support, and is bounded by DP. Thus, it could be a possible direction for modifying the standard risk notions that we consider to be more pessimistic, and is an interesting direction for future work.
>
> ____
>
> > Authors could add a more detailed description of the singling-out attack. It is somewhat difficult to parse from Definition 3.1, so some general description would make it easier to follow.
>
> In fact, we do include a more detailed description of singling out in Appendix B.2.
>
> ____
>
> Let us know if we can clarify anything else.

---

> > ### Comment · Reviewer_gnWx · 2025-08-05
> >
> > Thank you for your thorough response that addresses all my concerns. I'm happy to retain my original positive evaluation of the paper.

---

### Note · Authors · 2025-08-12

We would like to extend our gratitude for the detailed reviews and constructive feedback, and for the reviewers' and Area Chair's time.

The reviewer/author discussion phase, particularly the exchange between Reviewers GSiV and YNGr, helped us sharpen our paper's narrative. As a result of the discussion period, we have been able to better articulate the core technical contribution of using the $f$-DP framework for unified and tighter risk bounds, and to more clearly position our work within the broader context of privacy-risk evaluation and attack-aware calibration.

We are grateful for the opportunity to improve our work based on this feedback, and will incorporate all discussed revisions into the camera-ready version.

With best regards,

-- The Authors of Paper 18234

---

### Decision · Program_Chairs · 2025-09-17

**Decision:**

Accept (poster)

**Comment:**

The paper develops bounds via f-DP on the (marginal) success rate of various attacks. Empirically, the theoretical bounds are shown to be tighter than previous bounds on a number of applications. Reviewers found the unified approach interesting and useful, and the main reservations are about practicality/extensions of the strong adversary model and the types of attacks studied in the paper.